# Low-Rank Extragradient Method for Nonsmooth and Low-Rank Matrix Optimization Problems

**Dan Garber**
Technion - Israel Institute of Technology
Haifa, Israel 3200003
dangar@technion.ac.il

**Atara Kaplan**
Technion - Israel Institute of Technology
Haifa, Israel 3200003
ataragold@campus.technion.ac.il

## Abstract

Low-rank and nonsmooth matrix optimization problems capture many fundamental tasks in statistics and machine learning. While significant progress has been made in recent years in developing efficient methods for *smooth* low-rank optimization problems that avoid maintaining high-rank matrices and computing expensive high-rank SVDs, advances for nonsmooth problems have been slow paced.

In this paper we consider standard convex relaxations for such problems. Mainly, we prove that under a natural *generalized strict complementarity* condition and under the relatively mild assumption that the nonsmooth objective can be written as a maximum of smooth functions, the *extragradient method*, when initialized with a "warm-start" point, converges to an optimal solution with rate $O(1/t)$ while requiring only two *low-rank* SVDs per iteration. We give a precise trade-off between the rank of the SVDs required and the radius of the ball in which we need to initialize the method. We support our theoretical results with empirical experiments on several nonsmooth low-rank matrix recovery tasks, demonstrating that using simple initializations, the extragradient method produces exactly the same iterates when full-rank SVDs are replaced with SVDs of rank that matches the rank of the (low-rank) ground-truth matrix to be recovered.

## 1 Introduction

Low-rank and nonsmooth matrix optimization problems have many important applications in statistics, machine learning, and related fields, such as *sparse PCA* [21, 34], *robust PCA* [28, 33, 2, 9, 38], *phase synchronization* [41, 6, 29], *community detection and stochastic block models* [1][1], *low-rank and sparse covariance matrix recovery* [35], *robust matrix completion* [22, 10], and more. For many of these problems, convex relaxations, in which one replaces the nonconvex low-rank constraint with a trace-norm constraint, have been demonstrated in numerous papers to be highly effective both in theory (under suitable assumptions) and empirically (see references above). These convex relaxations can be formulated as the following general nonsmooth optimization problem:

$$\min_{\mathbf{X} \in \mathcal{S}_n} g(\mathbf{X}), \tag{1}$$

where $g : \mathbb{S}^n \to \mathbb{R}$ is convex but nonsmooth, and $\mathcal{S}_n = \{\mathbf{X} \in \mathbb{S}^n \mid \text{Tr}(\mathbf{X}) = 1, \ \mathbf{X} \succeq 0\}$ is the spectrahedron in $\mathbb{S}^n$, $\mathbb{S}^n$ being the space of $n \times n$ real symmetric matrices.

Problem (1), despite being convex, is notoriously difficult to solve in large scale. The simplest and most general approach applicable to it is the *projected subgradient method* [3, 7], which requires on

---

[1]in [41, 6, 29] and [1] the authors consider SDPs with linear objective function and affine constraints of the form $\mathcal{A}(\mathbf{X}) = \mathbf{b}$. By incorporating the linear constraints into the objective function via a $\ell_2$ penalty term of the form $\lambda\|\mathcal{A}(\mathbf{X}) - \mathbf{b}\|_2$, $\lambda > 0$, we obtain a nonsmooth objective function.

35th Conference on Neural Information Processing Systems (NeurIPS 2021).

each iteration to compute a Euclidean projection onto the spectrahedron $\mathcal{S}_n$, which in worst case amounts to $O(n^3)$ runtime per iteration. In many applications $g(\mathbf{X})$ follows a composite model, i.e., $g(\mathbf{X}) = h(\mathbf{X}) + w(\mathbf{X})$, where $h(\cdot)$ is convex and smooth and $w(\cdot)$ is convex and nonsmooth but admits a simple structure (e.g., nonsmooth regularizer). For such composite objectives, without the spectrahedron constraint, proximal methods such as FISTA [4] or splitting methods such as ADMM [32] are often very effective. However, with the spectrahderon constraint, all such methods require on each iteration to apply a subprocedure (e.g., computing the proximal mapping) which in worst case amounts to at least $O(n^3)$ runtime. A third type of off-the-shelf methods include those which are based on the *conditional gradient method* and adapted to nonsmooth problems, see for instance [31, 19, 36, 26]. The advantage of such methods is that no expensive high-rank SVD computations are needed. Instead, only a single leading eigenvector computation (i.e., a rank-one SVD) per iteration is required. However, whenever the number of iterations is not small, these methods still require to store high-rank matrices in memory, even when the optimal solution is low-rank. Thus, to conclude, standard first-order methods for Problem (1) require in worst case $\Omega(n^3)$ runtime per iteration and/or to store high-rank matrices in memory.

In the recent works [17, 18] it was established that for smooth objective functions, the high-rank SVD computations required for Euclidean projections onto the spectrahedron in standard gradient methods, can be replaced with low-rank SVDs in the close proximity of a low-rank optimal solution. This is significant since the runtime to compute a rank-$r$ SVD of a given $n \times n$ matrix using efficient iterative methods typically scales with $rn^2$ (and further improves when the matrix is sparse), instead of $n^3$ for a full-rank SVD. These results depend on the existence of eigen-gaps in the gradient of the optimal solution, which we refer to as a *generalized strict complementarity condition*. These results also hinge on a unique property of the Euclidean projection onto the spectrahedron. The projection onto the spectrahedron of a matrix $\mathbf{X} \in \mathbb{S}^n$, which admits an eigen-decomposition $\mathbf{X} = \sum_{i=1}^{n} \lambda_i \mathbf{v}_i \mathbf{v}_i^\top$, is given by

$$\Pi_{\mathcal{S}_n}[\mathbf{X}] = \sum_{i=1}^{n} \max\{0, \lambda_i - \lambda\} \mathbf{v}_i \mathbf{v}_i^\top, \qquad (2)$$

where $\lambda \in \mathbb{R}$ is the unique scalar satisfying $\sum_{i=1}^{n} \max\{0, \lambda_i - \lambda\} = 1$. This operation thus truncates all eigenvalues that are smaller than $\lambda$, while leaving the eigenvectors unchanged, thereby returning a matrix with rank equal to the number of eigenvalues greater than $\lambda$. Importantly, when the projection of $\mathbf{X}$ onto $\mathcal{S}_n$ is of rank $r$, only the first $r$ components in the eigen-decomposition of $\mathbf{X}$ are required to compute it in the first place, and thus, only a rank-$r$ SVD of $\mathbf{X}$ is required. In other words and simplifying, [17, 18] show that under strict complementary, at the proximity of an optimal solution of rank $r$, the exact Euclidean projection equals the rank-$r$ truncated projection given by:

$$\widehat{\Pi}_{\mathcal{S}_n}^r[\mathbf{X}] := \Pi_{\mathcal{S}_n}\left[\sum_{i=1}^{r} \lambda_i \mathbf{v}_i \mathbf{v}_i^\top\right]. \qquad (3)$$

Extending the results of [17, 18] to the nonsmooth setting is difficult since the smoothness assumption is critical to the analysis. Moreover, while [17, 18] rely on certain eigen-gaps in the gradients at optimal points, for nonsmooth problems, since the subdifferential set is often not a singleton, it is not likely that a similar eigen-gap property holds for all subgradients of an optimal solution.

In this paper we show that under the mild assumption that Problem (1) can be formulated as a smooth convex-concave saddle-point problem, i.e., the nonsmooth term can be written as a maximum over (possibly infinite number of) smooth convex functions, we can obtain results in the spirit of [17, 18]. Concretely, we show that if a generalized strict complementarity (GSC) assumption holds for a low-rank optimal solution (see Assumption 1 in the sequel), the *extragradient method* for smooth convex-concave saddle-point problems [23, 30] (see Algorithm 1 below), when initialized in the proximity of the optimal solution, converges with its original convergence rate of $O(1/t)$, while requiring only two low-rank SVDs per iteration[2]. It is important to recall that while the extragradient method requires two SVDs per iteration, it has the benefit of a fast $O(1/t)$ convergence rate, while simpler saddle-point methods such as mirror-descent-based only achieve a $O(1/\sqrt{t})$ rate [7].

Our contributions can be summarized as follows:

---

[2]note that the extradgradient method computes two projected-gradient steps on each iteration, and thus two SVDs are needed per iteration.

- We prove that even under (standard) strict complementarity, the projected subgradient method, when initialized with a "warm-start", may produce iterates with rank higher than that of the optimal solution. This phenomena further motivates our saddle-point approach. See Lemma 5.

- We suggest a generalized strict complementarity (GSC) condition for saddle-point problems and prove that when $g(\cdot)$ — the objective function in Problem (1), admits a highly popular saddle-point structure (one which captures all applications we mentioned in this paper), GSC w.r.t. an optimal solution to Problem (1) implies GSC (with the same parameters) w.r.t. a corresponding optimal solution of the equivalent saddle-point problem (the other direction always holds). See Section 3.

- **Main result:** we prove that for a smooth convex-concave saddle-point problem and an optimal solution which satisfies GSC, the extragradient method, when initialized with a "warm-start", converges with its original rate of $O(1/t)$ while requiring only two low-rank SVDs per iteration. Moreover, we prove GSC facilitates a precise and powerful tradeoff: increasing the rank of SVD computations (beyond the rank of the optimal solution) can significantly increase the radius of the ball in which the method needs to be initialized. See Theorem 1.

- We present extensive numerical evidence that demonstrate both the plausibility of the GSC assumption in various tasks, and more importantly, demonstrate that indeed the extragradient method with simple initialization converges correctly (i.e., produces exactly the same sequences of iterates) when the rank of the SVDs used to compute the (truncated) projections matches the rank of the (low-rank) ground-truth matrix to be recovered, instead of naively using full-rank SVDs (as suggested by (2)). See Section 5.

## 1.1 Additional related work

Since, as in the works [17, 18] mentioned before which deal with smooth objectives, strict complementarity plays a key role in our analysis, we refer the interested reader to the recent works [16, 39, 13, 20] which also exploit this property for efficient smooth and convex optimization over the spectrahedron. Strict complementarity has also played an instrumental role in two recent and very influential works which used it to prove linear convergence rates for proximal gradient methods [42, 14].

Besides convex relaxations such as Problem (1), considerable advances have been made in the past several yeas in developing efficient *nonconvex* methods with global convergence guarantees for low-rank matrix problems. In [37] the authors consider semidefinite programs and prove that under a smooth manifold assumption on the constraints, such methods converge to the optimal global solution. In [24] the authors prove global convergence of factorized nonconvex gradient descent from a "warm-start" initialization point for non-linear smooth minimization on the positive semidefinite cone. Very recently, [8] has established, under statistical conditions, fast convergence results from "warm-start" initialization of nonconvex first-order methods, when applied to nonsmooth nonconvex matrix recovery problems which are based on the explicit factorization of the low-rank matrix. A result of similar flavor concerning nonsmooth and nonconvex formulation of robust recovery of low-rank matrices from random linear measurements was presented in [25]. Finally, several recent works have considered nonconvex low-rank regularizers which result in nonconvex nonsmooth optimization problems, but guarantee convergence only to a stationary point [27, 40].

## 2 Strict complementarity for nonsmooth optimization and difficulty of applying low-rank projected subgradient steps

Our analysis of the nonsmooth Problem (1) naturally depends on certain subgradients of an optimal solution which, in many aspects, behave like the gradients of smooth functions. The existence of such a subgradient is guaranteed from the first-order optimality condition for constrained convex minimization problems:

**Lemma 1** (first-order optimality condition, see [3]). *Let $g : \mathbb{S}^n \to \mathbb{R}$ be a convex function. Then $\mathbf{X}^* \in \mathcal{S}_n$ minimizes $g$ over $\mathcal{S}_n$ if and only if there exists a subgradient $\mathbf{G}^* \in \partial g(\mathbf{X}^*)$ such that $\langle \mathbf{X} - \mathbf{X}^*, \mathbf{G}^* \rangle \geq 0$ for all $\mathbf{X} \in \mathcal{S}_n$.*

For some $\mathbf{G}^* \in \partial g(\mathbf{X}^*)$ which satisfies the first-order optimality condition for an optimal solution $\mathbf{X}^*$, if the multiplicity of the smallest eigenvalue equals $r^* = \mathrm{rank}(\mathbf{X}^*)$, then it can be shown that the optimal solution satisfies a strict complementarity assumption. The equivalence between a standard strict complementarity assumption on some low-rank optimal solution of a *smooth* optimization problem over the spectrahedron and an eigen-gap in the gradient of the optimal solution was established in [39]. We generalize this equivalence to also include nonsmooth problems. The proof follows similar arguments and is given in Appendix A.1.

**Definition 1** (strict complementarity). *An optimal solution $\mathbf{X}^* \in \mathcal{S}_n$ of rank $r^*$ for Problem* (1) *satisfies the strict complementarity assumption with parameter $\delta > 0$, if there exists an optimal solution of the dual problem[3] $(\mathbf{Z}^*, s^*) \in \mathbb{S}^n \times \mathbb{R}$ such that $\mathrm{rank}(\mathbf{Z}^*) = n - r^*$, and $\lambda_{n-r^*}(\mathbf{Z}^*) \geq \delta$.*

**Lemma 2.** *Let $\mathbf{X}^* \in \mathcal{S}_n$ be a rank-$r^*$ optimal solution to Problem* (1). *$\mathbf{X}^*$ satisfies the (standard) strict complementarity assumption with parameter $\delta > 0$ if and only if there exists a subgradient $\mathbf{G}^* \in \partial g(\mathbf{X}^*)$ such that $\langle \mathbf{X} - \mathbf{X}^*, \mathbf{G}^* \rangle \geq 0$ for all $\mathbf{X} \in \mathcal{S}_n$ and $\lambda_{n-r^*}(\mathbf{G}^*) - \lambda_n(\mathbf{G}^*) \geq \delta$.*

Throughout this paper we assume a weaker and more general assumption than strict complementarity, namely generalized strict complementarity (GSC), which we present now.

**Assumption 1** (generalized strict complementarity). *We say an optimal solution $\mathbf{X}^*$ to Problem* (1) *satisfies the generalized strict complementarity assumption with parameters $r, \delta$, if there exists a subgradient $\mathbf{G}^* \in \partial g(\mathbf{X}^*)$ such that $\langle \mathbf{X} - \mathbf{X}^*, \mathbf{G}^* \rangle \geq 0$ for all $\mathbf{X} \in \mathcal{S}_n$ and $\lambda_{n-r}(\mathbf{G}^*) - \lambda_n(\mathbf{G}^*) \geq \delta$.*

In [17] the author presents several characteristic properties of the gradient of the optimal solution in optimization problems over the spectrahedron. Using the existence of subgradients which satisfy the condition in Lemma 1, we can extend these properties also to the nonsmooth setting. The following lemma shows that GSC with parameters $(r, \delta)$ for some $\delta > 0$ (Assumption 1) is a sufficient condition for the optimal solution to be of rank at most $r$. The proof follows immediately from the proof of the analogous Lemma 7 in [17], by replacing the gradient of the optimal solution with a subgradient for which the first-order optimality condition holds.

**Lemma 3.** *Let $\mathbf{X}^*$ be an optimal solution to Problem* (1) *and write its eigen-decomposition as $\mathbf{X}^* = \sum_{i=1}^{r^*} \lambda_i \mathbf{v}_i \mathbf{v}_i^T$. Then, any subgradient $\mathbf{G}^* \in \partial g(\mathbf{X}^*)$ which satisfies $\langle \mathbf{X} - \mathbf{X}^*, \mathbf{G}^* \rangle \geq 0$ for all $\mathbf{X} \in \mathcal{S}_n$, admits an eigen-decomposition such that the set of vectors $\{\mathbf{v}_i\}_{i=1}^{r^*}$ is a set of leading eigenvectors of $(-\mathbf{G}^*)$ which corresponds to the eigenvalue $\lambda_1(-\mathbf{G}^*) = -\lambda_n(\mathbf{G}^*)$. Furthermore, there exists at least one such subgradient.*

One motivation for assuming (standard) strict complementarity (Assumption 1 with parameters $r = \mathrm{rank}(\mathbf{X}^*)$ and $\delta > 0$) is that it guarantees a certain notion of robustness of the problem to small perturbations in the parameters. It is well known (see for instance [3]) that a projected subgradient step from $\mathbf{X}^*$ with respect to a subgradient $\mathbf{G}^* \in \partial g(\mathbf{X}^*)$ for which the first-order optimality condition holds, returns the optimal solution $\mathbf{X}^*$ itself. This implies that $\mathrm{rank}\left(\Pi_{\mathcal{S}_n}[\mathbf{X}^* - \eta \mathbf{G}^*]\right) = \mathrm{rank}(\mathbf{X}^*)$ (here $\eta$ is the step-size). Without (standard) strict complementarity however, a small change in the parameters could result in a higher rank matrix. This is captured in the following lemma which is analogous to Lemma 3 in [18], where again the proof is straightforward from the proof in [18] by replacing the gradient of the optimal solution with a subgradient for which the first-order optimality condition holds.

**Lemma 4.** *Let $\mathbf{X}^*$ be an optimal solution of rank $r^*$ to Problem* (1). *Let $\mathbf{G}^* \in \partial g(\mathbf{X}^*)$ be a subgradient at $\mathbf{X}^*$ such that $\langle \mathbf{X} - \mathbf{X}^*, \mathbf{G}^* \rangle \geq 0$ for all $\mathbf{X} \in \mathcal{S}_n$. Then, $\lambda_{n-r^*}(\mathbf{G}^*) = \lambda_n(\mathbf{G}^*)$ if and only if for any arbitrarily small $\zeta > 0$ it holds that $\mathrm{rank}\left(\Pi_{(1+\zeta)\mathcal{S}_n}[\mathbf{X}^* - \eta \mathbf{G}^*]\right) > r^*$, where $\eta > 0$, $(1+\zeta)\mathcal{S}_n = \{(1+\zeta)\mathbf{X} \mid \mathbf{X} \in \mathcal{S}_n\}$, and $\Pi_{(1+\zeta)\mathcal{S}_n}[\cdot]$ denotes the Euclidean projection onto the set $(1+\zeta)\mathcal{S}_n$.*

## 2.1 The challenge of applying low-rank projected subgradient steps

We now demonstrate the difficulty of replacing the full-rank SVD computations required in projected subgradient steps over the spectrahedron, with their low-rank SVD counterparts when attempting to solve Problem (1). We prove that a projected subgradient step from a point arbitrarily close to a

---

[3] Denote $q(\mathbf{Z}, s) = \min_{\mathbf{X} \in \mathbb{S}^n} \{g(\mathbf{X}) + s(1 - \mathrm{Tr}(\mathbf{X})) - \langle \mathbf{Z}, \mathbf{X} \rangle\}$. The dual problem of Problem (1) can be written as: $\max_{\{\mathbf{Z} \succeq 0, \ s \in \mathbb{R}\}} \{q(\mathbf{Z}, s) \mid (\mathbf{Z}, s) \in \mathrm{dom}(q)\}$.

low-rank optimal solution — even one that satisfies strict complementarity (Definition 1), may result in a higher rank matrix. The problem on which we demonstrate this phenomena is a well known convex formulation of the *sparse PCA* problem [12]. The proof is given in Appendix A.2.

**Lemma 5** (failure of low-rank subgradient descent on sparse PCA)**.** *Consider the problem* $\min_{\mathbf{X} \in \mathcal{S}_n} \{g(\mathbf{X}) := -\langle \mathbf{z}\mathbf{z}^\top + \mathbf{z}_\perp \mathbf{z}_\perp^\top, \mathbf{X} \rangle + \frac{1}{2k}\|\mathbf{X}\|_1\}$, *where* $\mathbf{z} = (1/\sqrt{k}, \ldots, 1/\sqrt{k}, 0, \ldots, 0)^\top$ *is supported on the first $k$ entries,* $\mathbf{z}_\perp = (0, \ldots, 0, 1/\sqrt{n-k}, \ldots, 1/\sqrt{n-k})^\top$ *is supported on the last $n-k$ entries, and $k \le n/4$. Then,* $\mathbf{z}\mathbf{z}^\top$ *is a rank-one optimal solution for which strict complementarity holds. However, for any $\eta < \frac{2}{3}$ and any $\mathbf{v} \in \mathbb{R}^n$ such that $\|\mathbf{v}\| = 1$, support($\mathbf{v}$) $\subseteq$ support($\mathbf{z}$), and $\langle \mathbf{z}, \mathbf{v} \rangle^2 = 1 - \frac{1}{2}\|\mathbf{v}\mathbf{v}^\top - \mathbf{z}\mathbf{z}^\top\|_F^2 \ge 1 - \frac{1}{2k^2}$, it holds that rank $\left( \Pi_{\mathcal{S}_n}[\mathbf{v}\mathbf{v}^\top - \eta\mathbf{G}_{\mathbf{v}\mathbf{v}^\top}] \right) > 1$, where $\mathbf{G}_{\mathbf{v}\mathbf{v}^\top} = -\mathbf{z}\mathbf{z}^\top - \mathbf{z}_\perp \mathbf{z}_\perp^\top + \frac{1}{2k}sign(\mathbf{v}\mathbf{v}^\top) \in \partial g(\mathbf{v}\mathbf{v}^\top)$.*

Note that the subgradient of the $\ell_1$-norm which we choose for the projected subgradient step simply corresponds to the sign function, which is arguably the most natural choice.

## 3 From nonsmooth to saddle-point problems

To circumvent the difficulty demonstrated in Lemma 5 in incorporating low-rank SVDs into standard subgradient methods for solving Problem (1), we propose tackling the nonsmooth problem with saddle-point methods.

We assume the nonsmooth Problem (1) can be written as a maximum of smooth functions, i.e., $g(\mathbf{X}) = \max_{\mathbf{y} \in \mathcal{K}} f(\mathbf{X}, \mathbf{y})$, where $\mathcal{K} \subset \mathbb{Y}$ is some compact and convex subset of the finite linear space over the reals $\mathbb{Y}$ onto which it is efficient to compute Euclidean projections. We assume $f(\cdot, \mathbf{y})$ is convex for all $\mathbf{y} \in \mathcal{K}$ and $f(\mathbf{X}, \cdot)$ is concave for all $\mathbf{X} \in \mathcal{S}_n$. That is, we rewrite Problem (1) as the following equivalent saddle-point problem:

$$\min_{\mathbf{X} \in \mathcal{S}_n} \max_{\mathbf{y} \in \mathcal{K}} f(\mathbf{X}, \mathbf{y}). \tag{4}$$

Finding an optimal solution to problem (4) is equivalent to finding a saddle-point $(\mathbf{X}^*, \mathbf{y}^*) \in \mathcal{S}_n \times \mathcal{K}$ such that for all $\mathbf{X} \in \mathcal{S}_n$ and $\mathbf{y} \in \mathcal{K}$, $f(\mathbf{X}^*, \mathbf{y}) \le f(\mathbf{X}^*, \mathbf{y}^*) \le f(\mathbf{X}, \mathbf{y}^*)$.

We make a standard assumption that $f(\cdot, \cdot)$ is smooth with respect to all components. That is, we assume there exist $\beta_X, \beta_y, \beta_{Xy}, \beta_{yX} \ge 0$ such that for any $\mathbf{X}, \tilde{\mathbf{X}} \in \mathcal{S}_n$ and $\mathbf{y}, \tilde{\mathbf{y}} \in \mathcal{K}$ the following four inequalities hold: $\|\nabla_\mathbf{X} f(\mathbf{X}, \mathbf{y}) - \nabla_\mathbf{X} f(\tilde{\mathbf{X}}, \mathbf{y})\|_F \le \beta_X \|\mathbf{X} - \tilde{\mathbf{X}}\|_F$, $\|\nabla_\mathbf{y} f(\mathbf{X}, \mathbf{y}) - \nabla_\mathbf{y} f(\mathbf{X}, \tilde{\mathbf{y}})\|_2 \le \beta_y \|\mathbf{y} - \tilde{\mathbf{y}}\|_2$, $\|\nabla_\mathbf{X} f(\mathbf{X}, \mathbf{y}) - \nabla_\mathbf{X} f(\mathbf{X}, \tilde{\mathbf{y}})\|_F \le \beta_{Xy} \|\mathbf{y} - \tilde{\mathbf{y}}\|_2$, $\|\nabla_\mathbf{y} f(\mathbf{X}, \mathbf{y}) - \nabla_\mathbf{y} f(\tilde{\mathbf{X}}, \mathbf{y})\|_2 \le \beta_{yX} \|\mathbf{X} - \tilde{\mathbf{X}}\|_F$, where $\nabla_\mathbf{X} f = \frac{\partial f}{\partial \mathbf{X}}$ and $\nabla_\mathbf{y} f = \frac{\partial f}{\partial \mathbf{y}}$.

We denote by $\beta$ the full Lipschitz parameter of the gradient, that is for any $\mathbf{X}, \tilde{\mathbf{X}} \in \mathcal{S}_n$ and $\mathbf{y}, \tilde{\mathbf{y}} \in \mathcal{K}$,

$$\|(\nabla_\mathbf{X} f(\mathbf{X}, \mathbf{y}), -\nabla_\mathbf{y} f(\mathbf{X}, \mathbf{y})) - (\nabla_\mathbf{X} f(\tilde{\mathbf{X}}, \tilde{\mathbf{y}}), -\nabla_\mathbf{y} f(\tilde{\mathbf{X}}, \tilde{\mathbf{y}}))\| \le \beta \|(\mathbf{X}, \mathbf{Y}) - (\tilde{\mathbf{X}}, \tilde{\mathbf{y}})\|,$$

where $\|\cdot\|$ denotes the Euclidean norm over the product space $\mathcal{S}_n \times \mathbb{Y}$.

The relationship between the full Lipschitz parameter $\beta$ and its components $\beta_X, \beta_y, \beta_{Xy}, \beta_{yX}$ can be written as $\beta = \sqrt{2} \max \left\{ \sqrt{\beta_X^2 + \beta_{yX}^2}, \sqrt{\beta_y^2 + \beta_{Xy}^2} \right\}$. See proof in Appendix B.

The following lemma highlights a connection between the gradient of a saddle-point of (4) and subgradients of an optimal solution to (1) for which the first order optimality condition holds. One of the connections we will be interested in, is that GSC for Problem (1) implies GSC (with the same parameters) for Problem (4). However, to prove this specific connection we require an additional structural assumption on the objective function $g(\cdot)$. We note that this assumption holds for all applications mentioned in this paper.

**Assumption 2.** *$g(\mathbf{X})$ is of the form $g(\mathbf{X}) = h(\mathbf{X}) + \max_{\mathbf{y} \in \mathcal{K}} \mathbf{y}^\top (\mathcal{A}(\mathbf{X}) - \mathbf{b})$, where $h(\cdot)$ is smooth and convex, and $\mathcal{A}$ is a linear map.*

**Lemma 6.** *If $(\mathbf{X}^*, \mathbf{y}^*)$ is a saddle-point of Problem (4) then $\mathbf{X}^*$ is an optimal solution to Problem (1), $\nabla_\mathbf{X} f(\mathbf{X}^*, \mathbf{y}^*) \in \partial g(\mathbf{X}^*)$, and for all $\mathbf{X} \in \mathcal{S}_n$ it holds that $\langle \mathbf{X} - \mathbf{X}^*, \nabla_\mathbf{X} f(\mathbf{X}^*, \mathbf{y}^*) \rangle \ge 0$. Conversely, under Assumption 2, if $\mathbf{X}^*$ is an optimal solution to Problem (1), and $\mathbf{G}^* \in \partial g(\mathbf{X}^*)$ which satisfies $\langle \mathbf{X} - \mathbf{X}^*, \mathbf{G}^* \rangle \ge 0$ for all $\mathbf{X} \in \mathcal{S}_n$, then there exists $\mathbf{y}^* \in \arg\max_{\mathbf{y} \in \mathcal{K}} f(\mathbf{X}^*, \mathbf{y})$ such that $(\mathbf{X}^*, \mathbf{y}^*)$ is a saddle-point of Problem (4), and $\nabla_\mathbf{X} f(\mathbf{X}^*, \mathbf{y}^*) = \mathbf{G}^*$.*

The proof is given in Appendix C.1. The connection between the gradient of an optimal solution to the saddle-point problem and a subgradient of a corresponding optimal solution in the equivalent nonsmooth problem established in Lemma 6, naturally leads to the formulation of the following generalized strict complementarity assumption for saddle-point problems.

**Assumption 3** (generalized strict complementarity for saddle-points). *We say a saddle-point $(\mathbf{X}^*, \mathbf{y}^*) \in \mathcal{S}_n \times \mathcal{K}$ of Problem (4) with rank$(\mathbf{X}^*) = r^*$ satisfies the generalized strict complementarity assumption with parameters $r \geq r^*, \delta > 0$, if $\lambda_{n-r}(\nabla_{\mathbf{X}} f(\mathbf{X}^*, \mathbf{y}^*)) - \lambda_n(\nabla_{\mathbf{X}} f(\mathbf{X}^*, \mathbf{y}^*)) \geq \delta$.*

**Remark 1.** *Note that under Assumption 2, due to Lemma 6, GSC with parameters $r, \delta$ for some optimal solution $\mathbf{X}^*$ to Problem (1) implies GSC with parameters $r, \delta$ to a corresponding saddle-point $(\mathbf{X}^*, \mathbf{y}^*)$ of Problem (4). Nevertheless, Assumption 2 is not necessary for proving our convergence results for Problem (4), which are directly stated in terms of Assumption 3.*

## 4 Projected extragradient method with low-rank projections

In this section we formally state and prove our main result: the projected extragradient method for the saddle-point Problem (4), when initialized in the proximity of a saddle-point which satisfies GSC (Assumption 3), converges with its original $O(1/t)$ rate while requiring only two low-rank SVD computations per iteration.

---

**Algorithm 1** Projected extragradient method for saddle-point problems (see also [23, 30])

**Input:** sequence of step-sizes $\{\eta_t\}_{t \geq 1}$
**Initialization:** $(\mathbf{X}_1, \mathbf{y}_1) \in \mathcal{S}_n \times \mathcal{K}$
**for** $t = 1, 2, \dots$ **do**
$\quad \mathbf{Z}_{t+1} = \Pi_{\mathcal{S}_n}[\mathbf{X}_t - \eta_t \nabla_{\mathbf{X}} f(\mathbf{X}_t, \mathbf{y}_t)]$
$\quad \mathbf{w}_{t+1} = \Pi_{\mathcal{K}}[\mathbf{y}_t + \eta_t \nabla_{\mathbf{y}} f(\mathbf{X}_t, \mathbf{y}_t)]$
$\quad \mathbf{X}_{t+1} = \Pi_{\mathcal{S}_n}[\mathbf{X}_t - \eta_t \nabla_{\mathbf{X}} f(\mathbf{Z}_{t+1}, \mathbf{w}_{t+1})]$
$\quad \mathbf{y}_{t+1} = \Pi_{\mathcal{K}}[\mathbf{y}_t + \eta_t \nabla_{\mathbf{y}} f(\mathbf{Z}_{t+1}, \mathbf{w}_{t+1})]$
**end for**

---

**Theorem 1** (main theorem). *Fix an optimal solution $(\mathbf{X}^*, \mathbf{y}^*) \in \mathcal{S}_n \times \mathcal{K}$ to Problem (4). Let $\tilde{r}$ denote the multiplicity of $\lambda_n(\nabla_{\mathbf{X}} f(\mathbf{X}^*, \mathbf{y}^*))$ and for any $r \geq \tilde{r}$ define $\delta(r) = \lambda_{n-r}(\nabla_{\mathbf{X}} f(\mathbf{X}^*, \mathbf{y}^*)) - \lambda_n(\nabla_{\mathbf{X}} f(\mathbf{X}^*, \mathbf{y}^*))$. Let $\{(\mathbf{X}_t, \mathbf{y}_t)\}_{t \geq 1}$ and $\{(\mathbf{Z}_t, \mathbf{w}_t)\}_{t \geq 2}$ be the sequences of iterates generated by Algorithm 1 with a fixed step-size $\eta = \min\left\{ \frac{1}{2\sqrt{\beta_X^2 + \beta_{yX}^2}}, \frac{1}{2\sqrt{\beta_y^2 + \beta_{Xy}^2}}, \frac{1}{\beta_X + \beta_{Xy}}, \frac{1}{\beta_y + \beta_{yX}} \right\}$. Assume the initialization $(\mathbf{X}_1, \mathbf{y}_1)$ satisfies $\|(\mathbf{X}_1, \mathbf{y}_1) - (\mathbf{X}^*, \mathbf{y}^*)\| \leq R_0(r)$, where*

$$R_0(r) := \frac{\eta}{(1 + \sqrt{2})\left(1 + (2 + \sqrt{2})\eta \max\{\beta_X, \beta_{Xy}\}\right)} \max\left\{ \frac{\sqrt{\tilde{r}}\delta(r - \tilde{r} + 1)}{2}, \frac{\delta(r)}{(1 + 1/\sqrt{\tilde{r}})} \right\}.$$

*Then, for all $t \geq 1$, the projections $\Pi_{\mathcal{S}_n}[\mathbf{X}_t - \eta \nabla_{\mathbf{X}} f(\mathbf{X}_t, \mathbf{y}_t)]$ and $\Pi_{\mathcal{S}_n}[\mathbf{X}_t - \eta \nabla_{\mathbf{X}} f(\mathbf{Z}_{t+1}, \mathbf{w}_{t+1})]$ can be replaced with their rank-$r$ truncated counterparts (see (3)) without changing the sequences $\{(\mathbf{X}_t, \mathbf{y}_t)\}_{t \geq 1}$ and $\{(\mathbf{Z}_t, \mathbf{w}_t)\}_{t \geq 2}$, and for any $T \geq 0$ it holds that*

$$\frac{1}{T} \sum_{t=1}^{T} \max_{\mathbf{y} \in \mathcal{K}} f(\mathbf{Z}_{t+1}, \mathbf{y}) - \frac{1}{T} \sum_{t=1}^{T} \min_{\mathbf{X} \in \mathcal{S}_n} f(\mathbf{X}, \mathbf{w}_{t+1})$$

$$\leq \frac{D^2 \max\left\{ \sqrt{\beta_X^2 + \beta_{yX}^2}, \sqrt{\beta_y^2 + \beta_{Xy}^2}, \frac{1}{2}(\beta_X + \beta_{Xy}), \frac{1}{2}(\beta_y + \beta_{yX}) \right\}}{T},$$

*where $D := \sup_{(\mathbf{X}, \mathbf{y}), (\mathbf{Z}, \mathbf{w}) \in \mathcal{S}_n \times \mathcal{K}} \|(\mathbf{X}, \mathbf{y}) - (\mathbf{Z}, \mathbf{w})\|$.*

**Remark 2.** *Note that Theorem 1 implies that if standard strict complementarity holds for Problem (4), that is Assumption 3 holds with $r = r^* = \text{rank}(\mathbf{X}^*)$ and some $\delta > 0$, then only rank-$r^*$ SVDs are required so that Algorithm 1 converges with the guaranteed convergence rate of $O(1/t)$, when initialized with a "warm-start". Furthermore, by using SVDs of rank $r > r^*$, with moderately higher values of $r$, we can increase the radius of the ball in which Algorithm 1 needs to be initialized quite significantly.*

The complete proof of Theorem 1 is given in Appendix D. Below we give a short sketch of the proof.

*Proof sketch for Theorem 1.* The convergence rate of Algorithm 1 is well known in the literature (see Appendix D.1 for completeness). Thus, we focus on the novel part of the low-rank projections. For simplicity consider the case $r = \tilde{r}$. From (2) it can be deduced that for any $\mathbf{P} \in \mathbb{S}^n$ it holds that $\mathrm{rank}(\Pi_{\mathcal{S}_n}[\mathbf{P}]) \leq \tilde{r}$ if and only if the condition $\sum_{i=1}^{\tilde{r}} \lambda_i(\mathbf{P}) \geq 1 + \tilde{r}\lambda_{\tilde{r}+1}(\mathbf{P})$ holds. Denote $\mathbf{P}^* = \mathbf{X}^* - \eta\nabla_\mathbf{X} f(\mathbf{X}^*, \mathbf{y}^*)$. From Lemma 6 and Lemma 3 it follows that

$$\forall i \leq \mathrm{rank}(\mathbf{X}^*): \ \lambda_i(\mathbf{P}^*) = \lambda_i(\mathbf{X}^*) - \eta\lambda_n(\nabla_\mathbf{X} f(\mathbf{X}^*, \mathbf{y}^*));$$
$$\forall i > \mathrm{rank}(\mathbf{X}^*): \ \lambda_i(\mathbf{P}^*) = -\eta\lambda_{n-i+1}(\nabla_\mathbf{X} f(\mathbf{X}^*, \mathbf{y}^*)).$$

Using this and the fact that $\lambda_{n-i+1}(\nabla f_\mathbf{X}(\mathbf{X}^*, \mathbf{y}^*)) = \lambda_n(\nabla f_\mathbf{X}(\mathbf{X}^*, \mathbf{y}^*))$ for all $i \leq \tilde{r}$ we have,

$$\sum_{i=1}^{\tilde{r}} \lambda_i(\mathbf{P}^*) = \sum_{i=1}^{\mathrm{rank}(\mathbf{X}^*)} \lambda_i(\mathbf{X}^*) - \eta\sum_{i=1}^{\tilde{r}} \lambda_n(\nabla_\mathbf{X} f(\mathbf{X}^*, \mathbf{y}^*)) = \mathrm{Tr}(\mathbf{X}^*) - \eta\tilde{r}\lambda_n(\nabla_\mathbf{X} f(\mathbf{X}^*, \mathbf{y}^*))$$
$$= 1 - \eta\tilde{r}\lambda_{n-\tilde{r}}(\nabla_\mathbf{X} f(\mathbf{X}^*, \mathbf{y}^*)) + \eta\tilde{r}\delta(\tilde{r}) = 1 + \tilde{r}\lambda_{\tilde{r}+1}(\mathbf{P}^*) + \eta\tilde{r}\delta(\tilde{r}). \tag{5}$$

Thus, $\mathbf{P}^* = \mathbf{X}^* - \eta\nabla_\mathbf{X} f(\mathbf{X}^*, \mathbf{y}^*)$ not only satisfies the condition that ensures its projection onto $\mathcal{S}_n$ is of rank at most $\tilde{r}$, but it satisfies it with a positive slack of $\eta\tilde{r}\delta(\tilde{r})$. Also, for any $(\mathbf{X}, \mathbf{y})$ sufficiently close to $(\mathbf{X}^*, \mathbf{y}^*)$, using the smoothness of $f$, we have that $\mathbf{P} = \mathbf{X} - \eta\nabla_\mathbf{X} f(\mathbf{X}, \mathbf{y})$ is close to $\mathbf{P}^*$. Thus, by applying perturbation bounds for the eigenvalues of symmetric matrices to Eq. (5) and using the fact that the positive slack $\eta\tilde{r}\delta(\tilde{r})$ in the RHS of (5) allows to absorb sufficiently small errors, we can establish that for such points $(\mathbf{X}, \mathbf{y})$ it holds that $\mathrm{rank}(\Pi_{\mathcal{S}_n}[\mathbf{P}]) \leq \tilde{r}$. A similar argument shows that the second primal step in Algorithm 1: $\mathbf{X}_{t+1} = \Pi_{\mathcal{S}_n}[\mathbf{X}_t - \eta\nabla\mathbf{X} f(\mathbf{Z}_{t+1}, \mathbf{w}_{t+1})]$, also results in matrices of rank at most $\tilde{r}$ when $\mathbf{Z}_t, \mathbf{w}_t$ are also in the proximity of $\mathbf{X}^*, \mathbf{y}^*$. Finally, we prove a complementary argument, that when initialized in the proximity of $(\mathbf{X}^*, \mathbf{y}^*)$, the iterates of Algorithm 1 stay in the proximity of $(\mathbf{X}^*, \mathbf{y}^*)$ throughout all iterations.

□

**Remark 3.** *Note that when applying Algorithm 1 towards minimizing a nonsmooth objective of the form $g(\mathbf{X}) = \max_{\mathbf{y}\in\mathcal{K}} f(\mathbf{X}, \mathbf{y})$ (as discussed in Section 3), a guarantee of the form $\frac{1}{T}\sum_{t=1}^{T} \max_{\mathbf{y}\in\mathcal{K}} f(\mathbf{Z}_{t+1}, \mathbf{y}) - \frac{1}{T}\sum_{t=1}^{T} \min_{\mathbf{X}\in\mathcal{S}_n} f(\mathbf{X}, \mathbf{w}_{t+1}) \leq \epsilon$, for some $\epsilon > 0$, which is what we get from Theorem 1, implies in particular that $\min_{t\in[T]} g(\mathbf{X}_t) - g(\mathbf{X}^*) \leq \epsilon$, where $\mathbf{X}^*$ is a minimizer of $g(\cdot)$ over $\mathcal{S}_n$.*

**Remark 4.** *A downside of considering the saddle-point formulation* (4) *when attempting to solve Problem* (1) *that arises from Theorem 1, is that not only do we need a "warm-start" initialization for the original primal matrix variable $\mathbf{X}$, in the saddle-point formulation we need a "warm-start" for the primal-dual pair $(\mathbf{X}, \mathbf{y})$. Nevertheless, as we demonstrate extensively in Section 5, it seems that very simple initialization schemes work very well in practice.*

### 4.1 Efficiently-computable certificates for correctness of low-rank projections

Since Theorem 1 only applies in some neighborhood of an optimal solution, it is of interest to have a procedure for verifying if the rank-$r$ truncated projection of a given point indeed equals the exact Euclidean projection. In particular, from a practical point of view, it does not matter whether the conditions of Theorem 1 hold. In practice, as long as the truncated projection (see (3)) equals the exact projection (see (2)), we are guaranteed that Algorithm 1 converges correctly with rate $O(1/t)$, without needing to verify any other condition. Luckily, the expression in (2) which characterizes the structure of the Euclidean projection onto the spectrahedron, yields exactly such a verification procedure. As already noted in [17], for any $\mathbf{X} \in \mathbb{S}^n$, we have $\widehat{\Pi}_{\mathcal{S}_n}^r[\mathbf{X}] = \Pi_{\mathbf{S}_n}[\mathbf{X}]$ if an only if the condition $\sum_{i=1}^{r} \lambda_i(\mathbf{X}) \geq 1 + r \cdot \lambda_{r+1}(\mathbf{X})$ holds. Note that verifying this condition simply requires increasing the rank of the SVD computation by one, i.e., computing a rank-$(r+1)$ SVD of the matrix to project rather than a rank-$r$ SVD.

## 5 Empirical evidence

The goal of this section is to bring empirical evidence in support of our theoretical approach. We consider various tasks that take the form of minimizing a composite objective, i.e., the sum of a

smooth convex function and a nonsmooth convex function, where the nonsmoothness comes from either an $\ell_1$-norm or $\ell_2$-norm regularizer / penalty term, over a $\tau$-scaled spectrahedron. In all cases the nonsmooth objective can be written as a saddle-point with function $f(\mathbf{X}, \mathbf{y})$ which is linear in $\mathbf{y}$ and in particular satisfies Assumption 2.

The tasks considered include 1. sparse PCA, 2. robust PCA, 3. low-rank and sparse recovery, 4. phase synchronization, and 5. linearly-constrained low-rank estimation, under variety of parameters. Due to lack of space many of the results are deferred to Appendix F.

For all tasks considered we generate random instances, and examine the sequences of iterates generated by Algorithm 1 $\{(\mathbf{X}_t, \mathbf{y}_t)\}_{t \geq 1}$, $\{(\mathbf{Z}_t, \mathbf{w}_t)\}_{t \geq 2}$, when initialized with simple initialization procedures. Out of both sequences generated, we choose our candidate for the optimal solution to be the iterate for which the dual-gap, which is a certificate for optimality, is smallest. See Appendix E.

In all tasks considered the goal is to recover a ground-truth low-rank matrix $\mathbf{M}_0 \in \mathbb{S}^n$ from some noisy observation of it $\mathbf{M} = \mathbf{M}_0 + \mathbf{N}$, where $\mathbf{N} \in \mathbb{S}^n$ is a noise matrix. We measure the signal-to-noise ratio (SNR) as $\|\mathbf{M}_0\|_F^2 / \|\mathbf{N}\|_F^2$. In all experiments we measure the relative initialization error by $\left\| \frac{\text{Tr}(\mathbf{M}_0)}{\tau} \mathbf{X}_1 - \mathbf{M}_0 \right\|_F^2 / \|\mathbf{M}_0\|_F^2$, and similarly we measure the relative recovery error by $\left\| \frac{\text{Tr}(\mathbf{M}_0)}{\tau} \mathbf{X}^* - \mathbf{M}_0 \right\|_F^2 / \|\mathbf{M}_0\|_F^2$. Note that in some of the experiments we take $\tau < \text{Tr}(\mathbf{M}_0)$ to prevent the method from overfitting the noise. In addition, we measure the (standard) strict complementarity parameter which corresponds to the eigen-gap $\text{gap}(\nabla_{\mathbf{X}} f(\mathbf{X}^*, \mathbf{y}^*)) := \lambda_{n-r}(\nabla_{\mathbf{X}} f(\mathbf{X}^*, \mathbf{y}^*)) - \lambda_n(\nabla_{\mathbf{X}} f(\mathbf{X}^*, \mathbf{y}^*))$, $r = \text{rank}(\mathbf{M}_0)$.

In all experiments we use SVDs of rank $r = \text{rank}(\mathbf{M}_0)$ to compute the projections in Algorithm 1 according to the truncated projection given in (3). To certify the correctness of these low-rank projections (that is, that they equal the exact Euclidean projection) we confirm that the inequality $\sum_{i=1}^{r} \lambda_i(\mathbf{P}_j) \geq \tau + r \cdot \lambda_{r+1}(\mathbf{P}_j)$ always holds for $\mathbf{P}_1 = \mathbf{X}_t - \eta \nabla_{\mathbf{X}} f(\mathbf{X}_t, \mathbf{Y}_t)$ and $\mathbf{P}_2 = \mathbf{X}_t - \eta \nabla_{\mathbf{X}} f(\mathbf{Z}_{t+1}, \mathbf{W}_{t+1})$ (see also Section 4.1). Indeed, we can now already state our main observation from the experiments:

> *In all tasks considered and for all random instances generated, throughout all iterations of Algorithm 1, when initialized with a simple "warm-start" strategy and when computing only rank-$r$ truncated projections, $r = \text{rank}(\mathbf{M}_0)$, the truncated projections of $\mathbf{P}_1 = \mathbf{X}_t - \eta \nabla_{\mathbf{X}} f(\mathbf{X}_t, \mathbf{Y}_t)$ and $\mathbf{P}_2 = \mathbf{X}_t - \eta \nabla_{\mathbf{X}} f(\mathbf{Z}_{t+1}, \mathbf{W}_{t+1})$ equal their exact full-rank counterparts. That is, Algorithm 1, using only rank-$r$ SVDs, computed exactly the same sequences of iterates it would have computed if using full-rank SVDs.*

Aside from the above observation, in the sequel we demonstrate that all models considered indeed satisfy that: 1. the returned solution, denoted $(\mathbf{X}^*, \mathbf{y}^*)$, is of the same rank as the ground-truth matrix and satisfies the strict complementarity condition with non-negligible parameter (measured by the eigengap $\lambda_{n-r}(\nabla_{\mathbf{X}} f(\mathbf{X}^*, \mathbf{y}^*)) - \lambda_n(\nabla_{\mathbf{X}} f(\mathbf{X}^*, \mathbf{y}^*))$), 2. the recovery error of the returned solution indeed improves significantly over the error of the initialization point.

**Sparse PCA:** We consider the sparse PCA problem in a well known convex formulation taken from [12] and its equivalent saddle-point formulation:

$$\min_{\substack{\text{Tr}(\mathbf{X})=1, \\ \mathbf{X} \succeq 0}} \langle \mathbf{X}, -\mathbf{M} \rangle + \lambda \|\mathbf{X}\|_1 = \min_{\substack{\text{Tr}(\mathbf{X})=1, \\ \mathbf{X} \succeq 0}} \max_{\|\mathbf{Y}\|_\infty \leq 1} \{ \langle \mathbf{X}, -\mathbf{M} \rangle + \lambda \langle \mathbf{X}, \mathbf{Y} \rangle \},$$

where $\mathbf{M} = \mathbf{z}\mathbf{z}^\top + \frac{c}{2}(\mathbf{N} + \mathbf{N}^\top)$ is a noisy observation of a rank-one matrix $\mathbf{z}\mathbf{z}^\top$, with $\mathbf{z}$ being a sparse unit vector. Each entry $\mathbf{z}_i$ is chosen to be 0 with probability 0.9 and $U\{1, \ldots, 10\}$ with probability 0.1, and then we normalize $\mathbf{z}$ to be of unit norm. We test the results obtained when adding different magnitudes of Gaussian or uniform noise. We set the signal-to-noise ratio (SNR) to be a constant. Thus, we set the noise level to $c = \frac{2}{\text{SNR} \cdot \|\mathbf{N} + \mathbf{N}^\top\|_F}$ for our choice of SNR. We initialize the $\mathbf{X}$ variable with the rank-one approximation of $\mathbf{M}$. That is, we take $\mathbf{X}_1 = \mathbf{u}_1 \mathbf{u}_1^\top$, where $\mathbf{u}_1$ is the top eigenvector of $\mathbf{M}$. For the $\mathbf{Y}$ variable we initialize it with $\mathbf{Y}_1 = \text{sign}(\mathbf{X}_1)$ which is a subgradient of $\|\mathbf{X}_1\|_1$. We set the step-size to $\eta = 1/(2\lambda)$ and we set the number of iterations to $T = 1000$ and for any set of parameters we average the measurements over 10 i.i.d. runs.

Table 1: Numerical results for the sparse PCA problem. $\mathbf{N}_{ij} \sim U[0,1]$, SNR $= 0.05$.

| dimension (n) | 100 | 200 | 400 | 600 |
|---|---|---|---|---|
| $\lambda$ | 0.04 | 0.02 | 0.01 | 0.0067 |
| initialization error | 1.7456 | 1.7494 | 1.7566 | 1.7625 |
| recovery error | 0.0425 | 0.0244 | 0.0149 | 0.0100 |
| dual gap | $2.0 \times 10^{-9}$ | $5.8 \times 10^{-6}$ | $4.5 \times 10^{-4}$ | 0.0018 |
| gap($\nabla_{\mathbf{X}} f(\mathbf{X}^*, \mathbf{y}^*)$) | 0.7092 | 0.7854 | 0.8340 | 0.8622 |

**Robust PCA:** We consider the robust PCA problem [28] in the following formulation:

$$\min_{\substack{\text{Tr}(\mathbf{X})=\tau, \\ \mathbf{X} \succeq 0}} \|\mathbf{X} - \mathbf{M}\|_1 = \min_{\substack{\text{Tr}(\mathbf{X})=\tau, \\ \mathbf{X} \succeq 0}} \max_{\|\mathbf{Y}\|_\infty \leq 1} \langle \mathbf{X} - \mathbf{M}, \mathbf{Y} \rangle,$$

where $\mathbf{M} = r\mathbf{Z}_0\mathbf{Z}_0^\top + \frac{1}{2}(\mathbf{N} + \mathbf{N}^\top)$ is a sparsely-corrupted observation of some rank-r matrix $\mathbf{Z}_0\mathbf{Z}_0^\top$. We choose $\mathbf{Z}_0 \in \mathbb{R}^{n \times r}$ to be a random unit Frobenius norm matrix. For $\mathbf{N} \in \mathbb{R}^{n \times n}$, we choose each entry to be 0 with probability $1 - 1/\sqrt{n}$ and otherwise 1 or $-1$ with equal probability. We initialize the $\mathbf{X}$ variable with the projection $\mathbf{X}_1 = \Pi_{\{\text{Tr}(\mathbf{X})=\tau, \mathbf{x} \succeq 0\}}[\mathbf{M}]$, and the $\mathbf{Y}$ variable with $\mathbf{Y}_1 = \text{sign}(\mathbf{X}_1 - \mathbf{M})$. We vary the rank of $\mathbf{Z}_0$, which in turn determines the step-size $\eta$. We set the trace bound to $\tau = 0.95 \cdot \text{Tr}(r\mathbf{Z}_0\mathbf{Z}_0^\top)$, and the number of iterations to $T = 30,000$. For every set of parameters we average the measurements over 10 i.i.d. runs.

Table 2: Numerical results for the robust PCA problem. rank($\mathbf{Z}_0\mathbf{Z}_0^\top$) $= 10$, step-size $\eta = 1$.

| dimension (n) | 100 | 200 | 400 | 600 |
|---|---|---|---|---|
| SNR | 0.0229 | 0.0077 | 0.0026 | 0.0014 |
| initialization error | 1.5729 | 1.6485 | 1.6317 | 1.5949 |
| recovery error | 0.0079 | 0.0081 | 0.0073 | 0.0065 |
| dual gap | 0.0139 | 0.0338 | 0.1533 | 0.3561 |
| gap($\nabla_{\mathbf{X}} f(\mathbf{X}^*, \mathbf{y}^*)$) | 1.7945 | 16.9890 | 48.9799 | 82.2727 |

**Linearly-constrained low-rank estimation:** Consider the following penalized formulation:

$$\min_{\substack{\text{Tr}(\mathbf{X})=1, \\ \mathbf{X} \succeq 0}} \langle \mathbf{X}, -\mathbf{M} \rangle + \lambda \|\mathcal{A}(\mathbf{X}) - \mathbf{b}\|_2 = \min_{\substack{\text{Tr}(\mathbf{X})=1, \\ \mathbf{X} \succeq 0}} \max_{\|\mathbf{y}\|_2 \leq 1} \langle \mathbf{X}, -\mathbf{M} \rangle + \lambda \langle \mathcal{A}(\mathbf{X}) - \mathbf{b}, \mathbf{y} \rangle,$$

where $\mathbf{M} = \mathbf{z}_0\mathbf{z}_0^\top + \frac{c}{2}(\mathbf{N} + \mathbf{N}^\top)$ is a noisy observation of some rank-one matrix $\mathbf{z}_0\mathbf{z}_0^\top$ such that $\|\mathbf{z}_0\|_2 = 1$ and the noise matrix is chosen $\mathbf{N} \sim \mathcal{N}(0, \mathbf{I}_n)$. We take $\mathcal{A}(\mathbf{X}) = (\langle \mathbf{A}_1, \mathbf{X} \rangle, \ldots, \langle \mathbf{A}_m, \mathbf{X} \rangle)^\top$ with matrices $\mathbf{A}_1, \ldots, \mathbf{A}_m \in \mathbb{S}^n$ of the form $\mathbf{A}_i = \mathbf{v}_i\mathbf{v}_i^\top$ such that $\mathbf{v}_i \sim \mathcal{N}(0, 1)$. We take $\mathbf{b} \in \mathbb{R}^m$ such that $b_i = \langle \mathbf{A}_i, \mathbf{z}_0\mathbf{z}_0^\top \rangle$. We initialize the $\mathbf{X}$ variable with the rank-one approximation of $\mathbf{M}$. That is, we take $\mathbf{X}_1 = \mathbf{u}_1\mathbf{u}_1^\top$, where $\mathbf{u}_1$ is the top eigenvector of $\mathbf{M}$. The $\mathbf{y}$ variable is initialized with $\mathbf{y}_1 = (\mathcal{A}(\mathbf{X}_1) - \mathbf{b})/\|\mathcal{A}(\mathbf{X}_1) - \mathbf{b}\|_2$. We set the number of constraints to $m = n$, the penalty parameter to $\lambda = 2$, and the step-size to $\eta = 1/(2\lambda)$. We set the number of iterations in each experiment to $T = 2000$ and for each value of $n$ we average the measurements over 10 i.i.d. runs.

**Low-rank and sparse matrix recovery:** We consider the problem of recovering a simultaneously low-rank and sparse covariance matrix [35], which can be written as the following saddle-point optimization problem:

$$\min_{\substack{\text{Tr}(\mathbf{X})=1, \\ \mathbf{X} \succeq 0}} \frac{1}{2}\|\mathbf{X} - \mathbf{M}\|_F^2 + \lambda \|\mathbf{X}\|_1 = \min_{\substack{\text{Tr}(\mathbf{X})=\tau, \\ \mathbf{X} \succeq 0}} \max_{\|\mathbf{Y}\|_\infty \leq 1} \frac{1}{2}\|\mathbf{X} - \mathbf{M}\|_F^2 + \lambda \langle \mathbf{X}, \mathbf{Y} \rangle,$$

where $\mathbf{M} = \mathbf{Z}_0\mathbf{Z}_0^\top + \frac{c}{2}(\mathbf{N} + \mathbf{N}^\top)$ is a noisy observation of some low-rank and sparse covariance matrix $\mathbf{Z}_0\mathbf{Z}_0^\top$. We choose $\mathbf{Z}_0 \in \mathbb{R}^{n \times r}$ to be a sparse matrix where each entry $\mathbf{Z}_{0i,j}$ is chosen to be

Table 3: Numerical results for the linearly-constrained low-rank matrix estimation problem.

| dimension (n) | 100 | 200 | 400 | 600 |
|---|---|---|---|---|
| SNR | 0.15 | 0.075 | 0.04 | 0.027 |
| initialization error | 0.1219 | 0.1324 | 0.1242 | 0.1228 |
| recovery error | 0.0437 | 0.0617 | 0.0685 | 0.0735 |
| dual gap | $5.3 \times 10^{-11}$ | $5.0 \times 10^{-12}$ | $8.5 \times 10^{-12}$ | $2.3 \times 10^{-11}$ |
| $\text{gap}(\nabla_{\mathbf{X}} f(\mathbf{X}^*, \mathbf{y}^*))$ | 0.2941 | 0.3409 | 0.4690 | 0.5069 |
| $\|\mathcal{A}(\mathbf{X}^*) - \mathbf{b}\|_2$ | 0.0080 | 0.0082 | 0.0079 | 0.0073 |

0 with probability 0.9 and $U\{1, \ldots, 10\}$ with probability 0.1, and then we normalize $\mathbf{Z}_0$ to be of unit Frobenius norm. We choose $\mathbf{N} \sim \mathcal{N}(0.5, \mathbf{I}_n)$. We test the model with $\text{rank}(\mathbf{Z}_0\mathbf{Z}_0^\top) = 5, 10$. We set the signal-to-noise ratio (SNR) to be a constant and set the noise level to $c = \frac{2\|\mathbf{Z}_0\mathbf{Z}_0^\top\|_F}{\text{SNR} \cdot \|\mathbf{N}+\mathbf{N}^\top\|_F}$ for our choice of SNR. We initialize the $\mathbf{X}$ variable with the rank-r approximation of $\mathbf{M}$. That is, we take $\mathbf{X}_1 = \mathbf{U}_r \text{diag}\left(\Pi_{\Delta_{\tau,r}}[\text{diag}(-\Lambda_r)]\right)\mathbf{U}_r^\top$, where $\mathbf{U}_r\Lambda_r\mathbf{U}_r^\top$ is the rank-r eigen-decomposition of $\mathbf{M}$ and $\Delta_{\tau,r} = \{\mathbf{z} \in \mathbb{R}^r \mid \mathbf{z} \geq 0, \sum_{i=1}^r \mathbf{z}_i = \tau\}$ is the simplex of radius $\tau$ in $\mathbb{R}^r$. For the $\mathbf{Y}$ variable we initialize it with $\mathbf{Y}_1 = \text{sign}(\mathbf{X}_1)$ which is a subgradient of $\|\mathbf{X}_1\|_1$. We set the step-size to $\eta = 1$, $\tau = 0.7 \cdot \text{Tr}(\mathbf{Z}_0\mathbf{Z}_0^\top)$, and the number of iterations in each experiment to $T = 2000$. For each value of $r$ and $n$ we average the measurements over 10 i.i.d. runs.

Table 4: Numerical results for the low-rank and sparse matrix recovery problem.

| dimension (n) | 100 | 200 | 400 | 600 |
|---|---|---|---|---|
| $\downarrow r = \text{rank}(\mathbf{Z}_0\mathbf{Z}_0^\top) = 5, \text{SNR} = 2.4 \downarrow$ | | | | |
| $\lambda$ | 0.0012 | 0.0006 | 0.0003 | 0.0002 |
| initialization error | 0.2132 | 0.2103 | 0.1983 | 0.1907 |
| recovery error | 0.0641 | 0.0478 | 0.0349 | 0.0274 |
| dual gap | $9.0 \times 10^{-4}$ | $4.3 \times 10^{-4}$ | $1.4 \times 10^{-4}$ | $7.3 \times 10^{-5}$ |
| $\text{gap}(\nabla_{\mathbf{X}} f(\mathbf{X}^*, \mathbf{y}^*))$ | 0.0148 | 0.0200 | 0.0257 | 0.0277 |
| $\downarrow r = \text{rank}(\mathbf{Z}_0\mathbf{Z}_0^\top) = 10, \text{SNR} = 4.8 \downarrow$ | | | | |
| $\lambda$ | 0.0007 | 0.0004 | 0.0002 | 0.0001 |
| initialization error | 0.1855 | 0.1661 | 0.1527 | 0.1473 |
| recovery error | 0.0702 | 0.0403 | 0.0268 | 0.0356 |
| dual gap | $4.9 \times 10^{-4}$ | $6.6 \times 10^{-4}$ | $4.2 \times 10^{-4}$ | $3.4 \times 10^{-5}$ |
| $\text{gap}(\nabla_{\mathbf{X}} f(\mathbf{X}^*, \mathbf{y}^*))$ | 0.0072 | 0.0142 | 0.0187 | 0.0160 |

## 6  Discussion

This work expands upon a line of research that aims to harness the ability of convex relaxations to produce low-rank and high-quality solutions to important low-rank matrix optimization problems, while insisting on methods that, at least locally, store and manipulate only low-rank matrices. Focusing on the challenging case of nonsmooth objective functions and following our evidence for the difficulties of obtaining such a result for subgradient methods (Lemma 5), we consider tackling nonsmooth objectives via saddle-point formulations. We prove that indeed under a generalized strict complementarity condition, a state-of-the-art method for convex-concave saddle-point problems converges locally while storing and manipulating only low-rank matrices. Extensive experiments over several tasks demonstrate that our conceptual approach of utilizing low-rank projections for more efficient optimization is not only of theoretical merit, but indeed seems to work well in practice.

## Acknowledgements

This research was supported by the ISRAEL SCIENCE FOUNDATION (grant No. 1108/18).

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
