# A  Proofs omitted from Section 2

## A.1  Proof of Lemma 2

We first restate the lemma and then prove it.

**Lemma 7.** *Let $\mathbf{X}^* \in \mathcal{S}_n$ be a rank-$r^*$ optimal solution to Problem* (1)*. $\mathbf{X}^*$ satisfies the (standard) strict complementarity assumption with parameter $\delta > 0$ if and only if there exists a subgradient $\mathbf{G}^* \in \partial g(\mathbf{X}^*)$ such that $\langle \mathbf{X} - \mathbf{X}^*, \mathbf{G}^* \rangle \geq 0$ for all $\mathbf{X} \in \mathcal{S}_n$ and $\lambda_{n-r^*}(\mathbf{G}^*) - \lambda_n(\mathbf{G}^*) \geq \delta$.*

*Proof.* By Slater's condition strong duality holds for Problem (1). Therefore, the KKT conditions for Problem (1) hold for the optimal solution $\mathbf{X}^*$ and some optimal dual solution $(\mathbf{Z}^*, s^*)$. The Lagrangian of Problem (1) can be written as

$$\mathcal{L}(\mathbf{X}, \mathbf{Z}, s) = g(\mathbf{X}) + s(1 - \mathrm{Tr}(\mathbf{X})) - \langle \mathbf{Z}, \mathbf{X} \rangle.$$

Thus, using the generalized KKT conditions for nonsmooth optimization problems (see Theorem 6.1.1 in [11]), this implies that for the primal and dual optimal solutions

$$\mathbf{0} \in \partial g(\mathbf{X}^*) - \mathbf{Z}^* - s^* \mathbf{I},$$
$$\langle \mathbf{X}^*, \mathbf{Z}^* \rangle = 0,$$
$$\mathrm{Tr}(\mathbf{X}^*) = 1,$$
$$\mathbf{X}^*, \mathbf{Z}^* \succeq 0.$$

The generalized first order optimality condition for unconstrained minimization implies that there exists some $\mathbf{G}^* \in \partial g(\mathbf{X}^*)$ for which $\mathbf{0} = \mathbf{G}^* - \mathbf{Z}^* - s^* \mathbf{I}$. It remains to be shown that $\langle \mathbf{X} - \mathbf{X}^*, \mathbf{G}^* \rangle \geq 0$ for all $\mathbf{X} \in \mathcal{S}_n$.

The cone of positive semidefinite matrices is self-dual, that is $\mathbf{Z}^* \succeq 0$ if and only if $\langle \mathbf{X}, \mathbf{Z}^* \rangle \geq 0$ for all $\mathbf{X} \in \mathcal{S}_n$. Therefore, $\mathbf{Z}^* \succeq 0$ if and only if for all $\mathbf{X} \in \mathcal{S}_n$ it holds that

$$0 \leq \langle \mathbf{X}, \mathbf{Z}^* \rangle = \langle \mathbf{X}, \mathbf{Z}^* \rangle - \langle \mathbf{X}^*, \mathbf{Z}^* \rangle + s^* \langle \mathbf{X} - \mathbf{X}^*, \mathbf{I} \rangle = \langle \mathbf{X} - \mathbf{X}^*, \mathbf{Z}^* + s^* \mathbf{I} \rangle$$
$$= \langle \mathbf{X} - \mathbf{X}^*, \mathbf{G}^* \rangle$$

as desired. The first equality holds using the complementarity condition and the property that $\mathrm{Tr}(\mathbf{X}) = \mathrm{Tr}(\mathbf{X}^*) = 1$.

Using the equality $\mathbf{G}^* = \mathbf{Z}^* + s^*\mathbf{I}$ it holds that

$$\lambda_{n-r^*}(\mathbf{Z}^*) = \lambda_{n-r^*}(\mathbf{Z}^*) + s^*\mathbf{I} - \lambda_n(\mathbf{Z}^*) - s^*\mathbf{I} = \lambda_{n-r^*}(\mathbf{Z}^* + s^*\mathbf{I}) - \lambda_n(\mathbf{Z}^* + s^*\mathbf{I})$$
$$= \lambda_{n-r^*}(\mathbf{G}^*) - \lambda_n(\mathbf{G}^*).$$

Thus, $\mathbf{X}^*$ satisfies the strict complementarity assumption with parameter $\delta > 0$, i.e., $\lambda_{n-r^*}(\mathbf{Z}^*) \geq \delta$, if and only if $\lambda_{n-r^*}(\mathbf{G}^*) - \lambda_n(\mathbf{G}^*) \geq \delta$. $\qquad\square$

## A.2 Proof of Lemma 5

We first restate the lemma and then prove it.

**Lemma 8.** *Consider the problem* $\min_{\mathbf{X}\in\mathcal{S}_n}\{g(\mathbf{X}) := -\langle\mathbf{z}\mathbf{z}^\top + \mathbf{z}_\perp\mathbf{z}_\perp^\top, \mathbf{X}\rangle + \frac{1}{2k}\|\mathbf{X}\|_1\}$, *where* $\mathbf{z} = (1/\sqrt{k},\ldots,1/\sqrt{k},0,\ldots,0)^\top$ *is supported on the first $k$ entries,* $\mathbf{z}_\perp = (0,\ldots,0,1/\sqrt{n-k},\ldots,1/\sqrt{n-k})^\top$ *is supported on the last $n-k$ entries, and $k \leq n/4$. Then, $\mathbf{z}\mathbf{z}^\top$ is a rank-one optimal solution for which strict complementarity holds. However, for any $\eta < \frac{2}{3}$ and any $\mathbf{v} \in \mathbb{R}^n$ such that $\|\mathbf{v}\| = 1$, $\mathrm{support}(\mathbf{v}) \subseteq \mathrm{support}(\mathbf{z})$, and $\langle\mathbf{z},\mathbf{v}\rangle^2 = 1 - \frac{1}{2}\|\mathbf{v}\mathbf{v}^\top - \mathbf{z}\mathbf{z}^\top\|_F^2 \geq 1 - \frac{1}{2k^2}$, it holds that $\mathrm{rank}\left(\Pi_{\mathcal{S}_n}[\mathbf{v}\mathbf{v}^\top - \eta\mathbf{G}_{\mathbf{v}\mathbf{v}^\top}]\right) > 1$, where* $\mathbf{G}_{\mathbf{v}\mathbf{v}^\top} = -\mathbf{z}\mathbf{z}^\top - \mathbf{z}_\perp\mathbf{z}_\perp^\top + \frac{1}{2k}\mathrm{sign}(\mathbf{v}\mathbf{v}^\top) \in \partial g(\mathbf{v}\mathbf{v}^\top)$.

*Proof.* $\mathbf{z}\mathbf{z}^\top$ is a rank-one optimal solution for this problem since for the subgradient $k\mathbf{z}\mathbf{z}^\top + 2k\mathbf{z}_\perp\mathbf{z}_\perp^\top \in \partial(\|\mathbf{z}\mathbf{z}^\top\|_1)$ the first-order optimality condition holds. Indeed, for all $\mathbf{X} \in \mathcal{S}_n$

$$\langle\mathbf{X} - \mathbf{z}\mathbf{z}^\top, -\mathbf{z}\mathbf{z}^\top - \mathbf{z}_\perp\mathbf{z}_\perp^\top + \frac{1}{2}\mathbf{z}\mathbf{z}^\top + \mathbf{z}_\perp\mathbf{z}_\perp^\top\rangle = \langle\mathbf{X} - \mathbf{z}\mathbf{z}^\top, -\frac{1}{2}\mathbf{z}\mathbf{z}^\top\rangle \geq 0. \qquad (6)$$

For the subgradient $-\frac{1}{2}\mathbf{z}\mathbf{z}^\top \in \partial g(\mathbf{z}\mathbf{z}^\top)$ there is a gap $\lambda_{n-1}(-\frac{1}{2}\mathbf{z}\mathbf{z}^\top) - \lambda_n(-\frac{1}{2}\mathbf{z}\mathbf{z}^\top) = \frac{1}{2} > 0$, and as we showed in (6) the first order optimality condition holds for $-\frac{1}{2}\mathbf{z}\mathbf{z}^\top$. Thus, by Lemma 2 the optimal solution $\mathbf{z}\mathbf{z}^\top$ satisfies standard strict complementarity.

We will show that the projection onto the spectrahedron of a subgradient step from $\mathbf{v}\mathbf{v}^\top$ with respect to the natural subgradient of the $\ell_1$-norm $\mathrm{sign}(\mathbf{v}\mathbf{v}^\top) \in \partial(\|\mathbf{v}\mathbf{v}^\top\|_1)$ returns a rank-2 solution.

It holds that

$$1 - \frac{1}{2k^2} \leq \langle\mathbf{z}\mathbf{z}^\top, \mathbf{v}\mathbf{v}^\top\rangle = \frac{1}{2}\left(\|\mathbf{z}\mathbf{z}^\top\|_F^2 + \|\mathbf{v}\mathbf{v}^\top\|_F^2 - \|\mathbf{v}\mathbf{v}^\top - \mathbf{z}\mathbf{z}^\top\|_F^2\right) = 1 - \frac{1}{2}\|\mathbf{v}\mathbf{v}^\top - \mathbf{z}\mathbf{z}^\top\|_F^2,$$

and equivalently

$$\sum_{i=1}^k\sum_{j=1}^k\left(\frac{1}{k} - (\mathbf{v}\mathbf{v}^\top)_{ij}\right)^2 = \|\mathbf{v}\mathbf{v}^\top - \mathbf{z}\mathbf{z}^\top\|_F^2 \leq \frac{1}{k^2}.$$

Therefore, for every $i,j \in \{1,\ldots,k\}$ it holds that

$$\left|(\mathbf{v}\mathbf{v}^\top)_{ij} - \frac{1}{k}\right| \leq \frac{1}{k},$$

which implies that $0 \leq (\mathbf{v}\mathbf{v}^\top)_{ij} \leq \frac{2}{k}$. Therefore, $k\mathbf{z}\mathbf{z}^\top = \mathrm{sign}(\mathbf{v}\mathbf{v}^\top) \in \partial(\|\mathbf{v}\mathbf{v}^\top\|_1)$.

Taking a projected subgradient step from $\mathbf{v}\mathbf{v}^\top$ with respect to the subgradient $-\mathbf{z}\mathbf{z}^\top - \mathbf{z}_\perp\mathbf{z}_\perp^\top + \frac{1}{2}\mathbf{z}\mathbf{z}^\top \in \partial g(\mathbf{v}\mathbf{v}^\top)$ has the form

$$\Pi_{\mathcal{S}_n}\left[\mathbf{v}\mathbf{v}^\top - \eta\left(-\mathbf{z}\mathbf{z}^\top - \mathbf{z}_\perp\mathbf{z}_\perp^\top + \frac{1}{2}\mathbf{z}\mathbf{z}^\top\right)\right] = \Pi_{\mathcal{S}_n}\left[\mathbf{v}\mathbf{v}^\top + \frac{\eta}{2}\mathbf{z}\mathbf{z}^\top + \eta\mathbf{z}_\perp\mathbf{z}_\perp^\top\right].$$

Since $\mathrm{support}(\mathbf{v}) \subseteq \mathrm{support}(\mathbf{z})$ it holds that $\left(\mathbf{v}\mathbf{v}^\top + \frac{\eta}{2}\mathbf{z}\mathbf{z}^\top\right) \perp \mathbf{z}_\perp\mathbf{z}_\perp^\top$. $\mathbf{v}\mathbf{v}^\top + \frac{\eta}{2}\mathbf{z}\mathbf{z}^\top$ is a rank-2 matrix and so we can denote the eigen-decomposition of $\mathbf{v}\mathbf{v}^\top + \frac{\eta}{2}\mathbf{z}\mathbf{z}^\top + \eta\mathbf{z}_\perp\mathbf{z}_\perp^\top$ as $\mathbf{v}\mathbf{v}^\top + \frac{\eta}{2}\mathbf{z}\mathbf{z}^\top + \eta\mathbf{z}_\perp\mathbf{z}_\perp^\top = \lambda_1\mathbf{v}_1\mathbf{v}_1^\top + \lambda_2\mathbf{v}_2\mathbf{v}_2^\top + \eta\mathbf{z}_\perp\mathbf{z}_\perp^\top$, where $\lambda_1 \geq \lambda_2$. Thus, invoking (2) to calculate the projection we need to find the scalar $\lambda \in \mathbb{R}$ for which the following holds.

$$\max\{\lambda_1 - \lambda, 0\} + \max\{\lambda_2 - \lambda, 0\} + \max\{\eta - \lambda, 0\} + \sum_{i=4}^n\max\{0 - \lambda, 0\} = 1.$$

$\lambda_1$ is the largest eigenvalue of $\mathbf{v}\mathbf{v}^\top + \frac{\eta}{2}\mathbf{z}\mathbf{z}^\top + \eta\mathbf{z}_\perp\mathbf{z}_\perp^\top$ since, under our assumption that $\eta < 2/3$, we have that

$$\lambda_1 \geq \frac{1}{2}(\lambda_1 + \lambda_2) = \frac{1}{2}\text{Tr}(\mathbf{v}\mathbf{v}^\top + \frac{\eta}{2}\mathbf{z}\mathbf{z}^\top) = \frac{1}{2} + \frac{\eta}{4} > \eta.$$

Therefore, $\lambda < \lambda_1 \leq \lambda_1 + \lambda_2 = \text{Tr}(\mathbf{v}\mathbf{v}^\top + \frac{\eta}{2}\mathbf{z}\mathbf{z}^\top) = 1 + \frac{\eta}{2}$.

In addition, $\max\{\lambda_2, \eta\} \geq \eta$. Therefore,

$$\lambda_1 - \max\{\lambda_2, \eta\} \leq \lambda_1 + \lambda_2 - \eta = 1 + \frac{\eta}{2} - \eta < 1,$$

and so we must have that $\lambda < \max\{\lambda_2, \eta\} \leq \lambda_1$.

This implies that both $\max\{\lambda_1 - \lambda, 0\} > 0$ and $\max\{\max\{\lambda_2, \eta\} - \lambda, 0\} > 0$. Thus, using (2) we conclude that

$$\text{rank}\left(\Pi_{\mathcal{S}_n}\left[\mathbf{v}\mathbf{v}^\top - \eta\left(-\mathbf{z}\mathbf{z}^\top - \mathbf{z}_\perp\mathbf{z}_\perp^\top + \frac{1}{2}\mathbf{z}\mathbf{z}^\top\right)\right]\right) \geq 2.$$

$\square$

# B  Relationship between full Lipschitz parameter and its components

To establish the relationship between $\beta$ and $\beta_X, \beta_y, \beta_{Xy}, \beta_{yX}$, we can see that for all $\mathbf{X}, \tilde{\mathbf{X}} \in \mathcal{S}_n$ and all $\mathbf{y}, \tilde{\mathbf{y}} \in \mathcal{K}$ it holds that

$$\|(\nabla_{\mathbf{X}} f(\mathbf{X}, \mathbf{y}), -\nabla_{\mathbf{y}} f(\mathbf{X}, \mathbf{y})) - (\nabla_{\mathbf{X}} f(\tilde{\mathbf{X}}, \tilde{\mathbf{y}}), -\nabla_{\mathbf{y}} f(\tilde{\mathbf{X}}, \tilde{\mathbf{y}}))\|^2$$
$$= \|\nabla_{\mathbf{X}} f(\mathbf{X}, \mathbf{y}) - \nabla_{\mathbf{X}} f(\tilde{\mathbf{X}}, \mathbf{v})\|_F^2 + \|\nabla_{\mathbf{y}} f(\mathbf{X}, \mathbf{y}) - \nabla_{\mathbf{y}} f(\tilde{\mathbf{X}}, \tilde{\mathbf{y}})\|_2^2$$
$$\leq 2\|\nabla_{\mathbf{X}} f(\mathbf{X}, \mathbf{y}) - \nabla_{\mathbf{X}} f(\tilde{\mathbf{X}}, \mathbf{y})\|_F^2 + 2\|\nabla_{\mathbf{X}} f(\tilde{\mathbf{X}}, \mathbf{y}) - \nabla_{\mathbf{X}} f(\tilde{\mathbf{X}}, \tilde{\mathbf{y}})\|_F^2$$
$$\quad + 2\|\nabla_{\mathbf{y}} f(\mathbf{X}, \mathbf{y}) - \nabla_{\mathbf{y}} f(\tilde{\mathbf{X}}, \mathbf{y})\|_2^2 + 2\|\nabla_{\mathbf{y}} f(\tilde{\mathbf{X}}, \mathbf{y}) - \nabla_{\mathbf{y}} f(\tilde{\mathbf{X}}, \tilde{\mathbf{y}})\|_2^2$$
$$\leq 2(\beta_X^2 + \beta_{yX}^2)\|\mathbf{X} - \tilde{\mathbf{X}}\|_F^2 + 2(\beta_y^2 + \beta_{Xy}^2)\|\mathbf{y} - \tilde{\mathbf{y}}\|_F^2$$
$$\leq 2\max\{\beta_X^2 + \beta_{yX}^2, \beta_y^2 + \beta_{Xy}^2\}\|(\mathbf{X}, \mathbf{y}) - (\tilde{\mathbf{X}}, \tilde{\mathbf{y}})\|^2.$$

Therefore, $\beta = \sqrt{2}\max\left\{\sqrt{\beta_X^2 + \beta_{yX}^2}, \sqrt{\beta_y^2 + \beta_{Xy}^2}\right\}$.

# C  Proofs omitted from Section 3

## C.1  Proof of Lemma 6

We first restate the lemma and then prove it.

**Lemma 9.** *If* $(\mathbf{X}^*, \mathbf{y}^*)$ *is a saddle-point of Problem* (4) *then* $\mathbf{X}^*$ *is an optimal solution to Problem* (1), $\nabla_{\mathbf{X}} f(\mathbf{X}^*, \mathbf{y}^*) \in \partial g(\mathbf{X}^*)$, *and for all* $\mathbf{X} \in \mathcal{S}_n$ *it holds that* $\langle \mathbf{X} - \mathbf{X}^*, \nabla_{\mathbf{X}} f(\mathbf{X}^*, \mathbf{y}^*)\rangle \geq 0$. *Conversely, under Assumption 2, if* $\mathbf{X}^*$ *is an optimal solution to Problem* (1), *and* $\mathbf{G}^* \in \partial g(\mathbf{X}^*)$ *which satisfies* $\langle \mathbf{X} - \mathbf{X}^*, \mathbf{G}^*\rangle \geq 0$ *for all* $\mathbf{X} \in \mathcal{S}_n$, *then there exists* $\mathbf{y}^* \in \arg\max_{\mathbf{y} \in \mathcal{K}} f(\mathbf{X}^*, \mathbf{y})$ *such that* $(\mathbf{X}^*, \mathbf{y}^*)$ *is a saddle-point of Problem* (4), *and* $\nabla_{\mathbf{X}} f(\mathbf{X}^*, \mathbf{y}^*) = \mathbf{G}^*$.

*Proof.* For the first direction of the lemma, we first observe that for any $\mathbf{X}_1, \mathbf{X}_2 \in \mathcal{S}_n$ and $\tilde{\mathbf{y}}_1 \in \arg\max_{\mathbf{y} \in \mathcal{K}} f(\mathbf{X}_1, \mathbf{y})$, $\tilde{\mathbf{y}}_2 \in \arg\max_{\mathbf{y} \in \mathcal{K}} f(\mathbf{X}_2, \mathbf{y})$, using the gradient inequality for $f(\cdot, \tilde{\mathbf{y}}_2)$, it holds that

$$g(\mathbf{X}_1) = f(\mathbf{X}_1, \tilde{\mathbf{y}}_1) \geq f(\mathbf{X}_1, \tilde{\mathbf{y}}_2) \geq f(\mathbf{X}_2, \tilde{\mathbf{y}}_2) + \langle \nabla_{\mathbf{X}} f(\mathbf{X}_2, \tilde{\mathbf{y}}_2), \mathbf{X}_1 - \mathbf{X}_2\rangle$$
$$= g(\mathbf{X}_2) + \langle \nabla_{\mathbf{X}} f(\mathbf{X}_2, \tilde{\mathbf{y}}_2), \mathbf{X}_1 - \mathbf{X}_2\rangle.$$

Thus, $\nabla_{\mathbf{X}} f(\mathbf{X}_2, \tilde{\mathbf{y}}_2)$ is a subgradient of $g(\cdot)$ at $\mathbf{X}_2$, i.e., $\nabla_{\mathbf{X}} f(\mathbf{X}_2, \tilde{\mathbf{y}}_2) \in \partial g(\mathbf{X}_2)$.

In particular, for a saddle-point $(\mathbf{X}^*, \mathbf{y}^*) \in \mathcal{S}_n \times \mathcal{K}$ it holds that $\mathbf{y}^* \in \arg\max_{\mathbf{y} \in \mathcal{K}} f(\mathbf{X}^*, \mathbf{y})$, and therefore, it follows that $\nabla_{\mathbf{X}} f(\mathbf{X}^*, \mathbf{y}^*) \in \partial g(\mathbf{X}^*)$. In addition, for all $\mathbf{X} \in \mathcal{S}_n$ and $\tilde{\mathbf{y}} \in \arg\max_{\mathbf{y} \in \mathcal{K}} f(\mathbf{X}, \mathbf{y})$ we have

$$g(\mathbf{X}^*) = f(\mathbf{X}^*, \mathbf{y}^*) \leq f(\mathbf{X}, \mathbf{y}^*) \leq f(\mathbf{X}, \tilde{\mathbf{y}}) = g(\mathbf{X}),$$

which implies that $\mathbf{X}^*$ is an optimal solution to $\min_{\mathbf{X}\in\mathcal{S}_n} g(\mathbf{X})$.

Finally, we need to show that the subgradient $\nabla_{\mathbf{X}} f(\mathbf{X}^*, \mathbf{y}^*) \in \partial g(\mathbf{X}^*)$ indeed satisfies the first-order optimality condition for $g(\cdot)$ at $\mathbf{X}^*$. To see this, we observe that since $\mathbf{X}^*$ is an optimal solution to $\min_{\mathbf{X}\in\mathcal{S}_n} f(\mathbf{X}, \mathbf{y}^*)$, it follows from the first-order optimality condition for the problem $\min_{\mathbf{X}\in\mathcal{S}_n} f(\mathbf{X}, \mathbf{y}^*)$, that for all $\mathbf{W} \in \mathcal{S}_n$

$$\langle \mathbf{W} - \mathbf{X}^*, \nabla_{\mathbf{X}} f(\mathbf{X}^*, \mathbf{y}^*) \rangle \geq 0,$$

as needed.

For the second direction, let $\mathbf{X}^* \in \arg\min_{\mathbf{X}\in\mathcal{S}_n} g(\mathbf{X})$ and let $\mathbf{G}^* \in \partial g(\mathbf{X}^*)$ such that $\langle \mathbf{X} - \mathbf{X}^*, \mathbf{G}^* \rangle \geq 0$ for all $\mathbf{X} \in \mathcal{S}_n$. By Assumption 2 and using Danskin's theorem (see for instance [5]), the subdifferential set of $g(\mathbf{X}^*) = h(\mathbf{X}^*) + \max_{\mathbf{y}\in\mathcal{K}} \mathbf{y}^\top(\mathcal{A}(\mathbf{X}^*) - \mathbf{b})$ can be written as

$$\partial g(\mathbf{X}^*) = \nabla h(\mathbf{X}^*) + \text{conv}\left\{ \mathcal{A}^\top(\mathbf{y}) \,\Big|\, \mathbf{y} \in \arg\max_{\mathbf{y}\in\mathcal{K}} \mathbf{y}^\top(\mathcal{A}(\mathbf{X}^*) - \mathbf{b}) \right\}$$

$$= \nabla h(\mathbf{X}^*) + \mathcal{A}^\top\left( \text{conv}\left\{ \mathbf{y} \,\Big|\, \mathbf{y} \in \arg\max_{\mathbf{y}\in\mathcal{K}} \mathbf{y}^\top(\mathcal{A}(\mathbf{X}^*) - \mathbf{b}) \right\} \right)$$

$$= \nabla h(\mathbf{X}^*) + \mathcal{A}^\top\left( \left\{ \mathbf{y} \,\Big|\, \mathbf{y} \in \arg\max_{\mathbf{y}\in\mathcal{K}} \mathbf{y}^\top(\mathcal{A}(\mathbf{X}^*) - \mathbf{b}) \right\} \right)$$

$$= \nabla h(\mathbf{X}^*) + \mathcal{A}^\top\left( \left\{ \mathbf{y} \,\Big|\, \mathbf{y} \in \arg\max_{\mathbf{y}\in\mathcal{K}} f(\mathbf{X}^*, \mathbf{y}) \right\} \right),$$

where $\text{conv}\{\cdot\}$ denotes the convex hull operation and the third equality follows from the convexity of $\mathcal{K}$.

Thus, there exists some $\mathbf{y}^* \in \arg\max_{\mathbf{y}\in\mathcal{K}} f(\mathbf{X}^*, \mathbf{y})$ such that $\mathbf{G}^* = \nabla h(\mathbf{X}^*) + \mathcal{A}^\top(\mathbf{y}^*) = \nabla_{\mathbf{X}} f(\mathbf{X}^*, \mathbf{y}^*)$.

Since $\mathbf{y}^* \in \arg\max_{\mathbf{y}\in\mathcal{K}} f(\mathbf{X}^*, \mathbf{y})$, it follows that for all $\mathbf{y} \in \mathcal{K}$, $f(\mathbf{X}^*, \mathbf{y}^*) \geq f(\mathbf{X}^*, \mathbf{y})$. In addition, using the fact that $\mathbf{G}^*$ satisfies the first-order optimality condition, and using gradient inequality w.r.t. $f(\cdot, \mathbf{y}^*)$, we have that for all $\mathbf{X} \in \mathcal{S}_n$,

$$0 \leq \langle \mathbf{X} - \mathbf{X}^*, \mathbf{G}^* \rangle = \langle \mathbf{X} - \mathbf{X}^*, \nabla_{\mathbf{X}} f(\mathbf{X}^*, \mathbf{y}^*) \rangle \leq f(\mathbf{X}, \mathbf{y}^*) - f(\mathbf{X}^*, \mathbf{y}^*).$$

Thus, it follows that $f(\mathbf{X}, \mathbf{y}^*) \geq f(\mathbf{X}^*, \mathbf{y}^*)$. Therefore, $(\mathbf{X}^*, \mathbf{y}^*)$ is indeed a saddle-point of $f$.

$\square$

# D   Proofs omitted from Section 4

## D.1   Convergence rate of the extragradient method

The $O(1/t)$ convergence rate of the projected extragradient method is a well known result[4]. We present it here for completeness.

**Lemma 10.** *Let $\{(\mathbf{X}_t, \mathbf{y}_t)\}_{t\geq 1}$ and $\{(\mathbf{Z}_t, \mathbf{w}_t)\}_{t\geq 2}$ be the sequences generated by Algorithm 1 with a fixed step-size $\eta_t = \eta \leq \min\left\{ \frac{1}{\beta_X + \beta_{Xy}}, \frac{1}{\beta_y + \beta_{yX}}, \frac{1}{\beta_X + \beta_{yX}}, \frac{1}{\beta_y + \beta_{Xy}} \right\}$ then*

$$\frac{1}{T}\sum_{t=1}^{T} \max_{\mathbf{y}\in\mathcal{K}} f(\mathbf{Z}_{t+1}, \mathbf{y}) - \frac{1}{T}\sum_{t=1}^{T} \min_{\mathbf{X}\in\mathcal{S}_n} f(\mathbf{X}, \mathbf{w}_{t+1}) \leq \frac{D^2}{2\eta T},$$

*where $D := \sup_{(\mathbf{X},\mathbf{y}),(\tilde{\mathbf{X}},\tilde{\mathbf{y}})\in\mathcal{S}_n\times\mathcal{K}} \|(\mathbf{X}, \mathbf{y}) - (\tilde{\mathbf{X}}, \tilde{\mathbf{y}})\|$. In particular,*

1. *$\min_{t\in[T]} \left( \max_{\mathbf{y}\in\mathcal{K}} f(\mathbf{Z}_{t+1}, \mathbf{y}) - \min_{\mathbf{X}\in\mathcal{S}_n} f(\mathbf{X}, \mathbf{w}_{t+1}) \right) \leq \frac{D^2}{2\eta T}$.*

2. *$\max_{\mathbf{y}\in\mathcal{K}} f\left( \frac{1}{T}\sum_{t=1}^{T} \mathbf{Z}_{t+1}, \mathbf{y} \right) - \min_{\mathbf{X}\in\mathcal{S}_n} f\left( \mathbf{X}, \frac{1}{T}\sum_{t=1}^{T} \mathbf{w}_{t+1} \right) \leq \frac{D^2}{2\eta T}$.*

---

[4][30] prove this result with respect to the ergodic series. A small change in the proof allows us to prove the same with respect to the minimum and maximum iterates.

*Proof.* The projection theorem states that projecting some point $\mathbf{s}$ onto some closed and convex set $\mathcal{C}$ satisfies that for all $\mathbf{z} \in \mathcal{C}$ it holds that $\langle \Pi_\mathcal{C}[\mathbf{s}] - \mathbf{s}, \Pi_\mathcal{C}[\mathbf{s}] - \mathbf{z} \rangle \leq 0$. In particular, for any $\mathbf{X} \in \mathcal{S}_n$, using the updates for $\mathbf{X}_{t+1}$ and $\mathbf{Z}_{t+1}$, the two following inequalities hold:

$$\eta_t \langle \mathbf{Z}_{t+1} - \mathbf{X}, \nabla_\mathbf{X} f(\mathbf{X}_t, \mathbf{y}_t) \rangle \leq \langle \mathbf{X}_t - \mathbf{Z}_{t+1}, \mathbf{Z}_{t+1} - \mathbf{X} \rangle \tag{7}$$

$$\eta_t \langle \mathbf{X}_{t+1} - \mathbf{X}, \nabla_\mathbf{X} f(\mathbf{Z}_{t+1}, \mathbf{w}_{t+1}) \rangle \leq \langle \mathbf{X}_t - \mathbf{X}_{t+1}, \mathbf{X}_{t+1} - \mathbf{X} \rangle. \tag{8}$$

By the gradient inequality, for any $\mathbf{X} \in \mathcal{S}_n$

$$
\begin{aligned}
f(\mathbf{Z}_{t+1}, \mathbf{w}_{t+1}) - f(\mathbf{X}, \mathbf{w}_{t+1}) &\leq \langle \mathbf{Z}_{t+1} - \mathbf{X}, \nabla_\mathbf{X} f(\mathbf{Z}_{t+1}, \mathbf{w}_{t+1}) \rangle \\
&= \langle \mathbf{X}_{t+1} - \mathbf{X}, \nabla_\mathbf{X} f(\mathbf{Z}_{t+1}, \mathbf{w}_{t+1}) \rangle + \langle \mathbf{Z}_{t+1} - \mathbf{X}_{t+1}, \nabla_\mathbf{X} f(\mathbf{X}_t, \mathbf{y}_t) \rangle \\
&\quad + \langle \mathbf{Z}_{t+1} - \mathbf{X}_{t+1}, \nabla_\mathbf{X} f(\mathbf{Z}_{t+1}, \mathbf{w}_{t+1}) - \nabla_\mathbf{X} f(\mathbf{X}_t, \mathbf{y}_t) \rangle. \tag{9}
\end{aligned}
$$

We will bound these three terms separately.

For the first term, using (8) and the Pythagoras identity

$$
\begin{aligned}
\langle \mathbf{X}_{t+1} - \mathbf{X}, \nabla_\mathbf{X} f(\mathbf{Z}_{t+1}, \mathbf{w}_{t+1}) \rangle &\leq \frac{1}{\eta_t} \langle \mathbf{X}_t - \mathbf{X}_{t+1}, \mathbf{X}_{t+1} - \mathbf{X} \rangle \\
&= -\frac{1}{2\eta_t} \|\mathbf{X}_t - \mathbf{X}_{t+1}\|_F^2 + \frac{1}{2\eta_t} \|\mathbf{X}_t - \mathbf{X}\|_F^2 - \frac{1}{2\eta_t} \|\mathbf{X}_{t+1} - \mathbf{X}\|_F^2. \tag{10}
\end{aligned}
$$

For the second term, using (7) with $\mathbf{X} = \mathbf{X}_{t+1}$ and the Pythagoras identity

$$
\begin{aligned}
&\langle \mathbf{Z}_{t+1} - \mathbf{X}_{t+1}, \nabla_\mathbf{X} f(\mathbf{X}_t, \mathbf{y}_t) \rangle \\
&\leq \frac{1}{\eta_t} \langle \mathbf{X}_t - \mathbf{Z}_{t+1}, \mathbf{Z}_{t+1} - \mathbf{X}_{t+1} \rangle \\
&= -\frac{1}{2\eta_t} \|\mathbf{X}_t - \mathbf{Z}_{t+1}\|_F^2 + \frac{1}{2\eta_t} \|\mathbf{X}_t - \mathbf{X}_{t+1}\|_F^2 - \frac{1}{2\eta_t} \|\mathbf{Z}_{t+1} - \mathbf{X}_{t+1}\|_F^2. \tag{11}
\end{aligned}
$$

For the third term, using the Cauchy–Schwarz inequality, the $\beta_X$ and $\beta_{Xy}$ smoothness, and the inequality $2ab \leq a^2 + b^2$ we obtain

$$
\begin{aligned}
&\langle \mathbf{Z}_{t+1} - \mathbf{X}_{t+1}, \nabla_\mathbf{X} f(\mathbf{Z}_{t+1}, \mathbf{w}_{t+1}) - \nabla_\mathbf{X} f(\mathbf{X}_t, \mathbf{y}_t) \rangle \\
&\leq \|\mathbf{Z}_{t+1} - \mathbf{X}_{t+1}\|_F \cdot \|\nabla_\mathbf{X} f(\mathbf{Z}_{t+1}, \mathbf{w}_{t+1}) - \nabla_\mathbf{X} f(\mathbf{X}_t, \mathbf{y}_t)\|_F \\
&\leq \|\mathbf{Z}_{t+1} - \mathbf{X}_{t+1}\|_F \cdot \|\nabla_\mathbf{X} f(\mathbf{Z}_{t+1}, \mathbf{w}_{t+1}) - \nabla_\mathbf{X} f(\mathbf{X}_t, \mathbf{w}_{t+1})\|_F \\
&\quad + \|\mathbf{Z}_{t+1} - \mathbf{X}_{t+1}\|_F \cdot \|\nabla_\mathbf{X} f(\mathbf{X}_t, \mathbf{w}_{t+1}) - \nabla_\mathbf{X} f(\mathbf{X}_t, \mathbf{y}_t)\|_F \\
&\leq (\beta_X \|\mathbf{Z}_{t+1} - \mathbf{X}_t\|_F + \beta_{Xy} \|\mathbf{w}_{t+1} - \mathbf{y}_t\|_2) \cdot \|\mathbf{Z}_{t+1} - \mathbf{X}_{t+1}\|_F \\
&\leq \frac{\beta_X}{2} \left( \|\mathbf{Z}_{t+1} - \mathbf{X}_t\|_F^2 + \|\mathbf{Z}_{t+1} - \mathbf{X}_{t+1}\|_F^2 \right) + \frac{\beta_{Xy}}{2} \left( \|\mathbf{w}_{t+1} - \mathbf{y}_t\|_2^2 + \|\mathbf{Z}_{t+1} - \mathbf{X}_{t+1}\|_F^2 \right). \tag{12}
\end{aligned}
$$

Plugging (10), (11), and (12) into (9) we obtain

$$
\begin{aligned}
&f(\mathbf{Z}_{t+1}, \mathbf{w}_{t+1}) - f(\mathbf{X}, \mathbf{w}_{t+1}) \\
&\leq \frac{1}{2\eta_t} \left( \|\mathbf{X}_t - \mathbf{X}\|_F^2 - \|\mathbf{X}_{t+1} - \mathbf{X}\|_F^2 \right) + \left( \frac{\beta_X}{2} - \frac{1}{2\eta_t} \right) \|\mathbf{Z}_{t+1} - \mathbf{X}_t\|_F^2 \\
&\quad + \left( \frac{\beta_X + \beta_{Xy}}{2} - \frac{1}{2\eta_t} \right) \|\mathbf{Z}_{t+1} - \mathbf{X}_{t+1}\|_F^2 + \frac{\beta_{Xy}}{2} \|\mathbf{w}_{t+1} - \mathbf{y}_t\|_2^2.
\end{aligned}
$$

Using similar arguments, for any $\mathbf{y} \in \mathcal{K}$

$$
\begin{aligned}
f(\mathbf{Z}_{t+1}, \mathbf{y}) - f(\mathbf{Z}_{t+1}, \mathbf{w}_{t+1}) &\leq \frac{1}{2\eta_t} \left( \|\mathbf{y}_t - \mathbf{y}\|_2^2 - \|\mathbf{y}_{t+1} - \mathbf{y}\|_2^2 \right) + \left( \frac{\beta_y}{2} - \frac{1}{2\eta_t} \right) \|\mathbf{w}_{t+1} - \mathbf{y}_t\|_2^2 \\
&\quad + \left( \frac{\beta_y + \beta_{yX}}{2} - \frac{1}{2\eta_t} \right) \|\mathbf{w}_{t+1} - \mathbf{y}_{t+1}\|_2^2 + \frac{\beta_{yX}}{2} \|\mathbf{Z}_{t+1} - \mathbf{X}_t\|_F^2.
\end{aligned}
$$

Summing the last two inequalities, we obtain for $\eta_t \leq \min\left\{\frac{1}{\beta_X + \beta_{Xy}}, \frac{1}{\beta_y + \beta_{yX}}, \frac{1}{\beta_X + \beta_{yX}}, \frac{1}{\beta_y + \beta_{Xy}}\right\}$

$$f(\mathbf{Z}_{t+1}, \mathbf{y}) - f(\mathbf{X}, \mathbf{w}_{t+1}) \leq \frac{1}{2\eta_t}\left(\|(\mathbf{X}_t, \mathbf{y}_t) - (\mathbf{X}, \mathbf{y})\|^2 - \|(\mathbf{X}_{t+1}, \mathbf{y}_{t+1}) - (\mathbf{X}, \mathbf{y})\|^2\right)$$
$$+ \left(\frac{\beta_X + \beta_{yX}}{2} - \frac{1}{2\eta_t}\right)\|\mathbf{Z}_{t+1} - \mathbf{X}_t\|_F^2$$
$$+ \left(\frac{\beta_y + \beta_{Xy}}{2} - \frac{1}{2\eta_t}\right)\|\mathbf{w}_{t+1} - \mathbf{y}_t\|_2^2$$
$$+ \left(\frac{\beta_X + \beta_{Xy}}{2} - \frac{1}{2\eta_t}\right)\|\mathbf{Z}_{t+1} - \mathbf{X}_{t+1}\|_F^2$$
$$+ \left(\frac{\beta_y + \beta_{yX}}{2} - \frac{1}{2\eta_t}\right)\|\mathbf{w}_{t+1} - \mathbf{y}_{t+1}\|_2^2$$
$$\leq \frac{1}{2\eta_t}\left(\|(\mathbf{X}_t, \mathbf{y}_t) - (\mathbf{X}, \mathbf{y})\|^2 - \|(\mathbf{X}_{t+1}, \mathbf{y}_{t+1}) - (\mathbf{X}, \mathbf{y})\|^2\right).$$

Taking the maximum over all $\mathbf{y} \in \mathcal{K}$ and minimum over all $\mathbf{X} \in \mathcal{S}_n$, averaging over $t = 1, \ldots, T$, and taking a $\eta_t = \eta$

$$\frac{1}{T}\sum_{t=1}^{T}\max_{\mathbf{y}\in\mathcal{K}} f(\mathbf{Z}_{t+1}, \mathbf{y}) - \frac{1}{T}\sum_{t=1}^{T}\min_{\mathbf{X}\in\mathcal{S}_n} f(\mathbf{X}, \mathbf{w}_{t+1}) \leq \frac{1}{2\eta T}\|(\mathbf{X}_1, \mathbf{y}_1) - (\mathbf{X}, \mathbf{y})\|^2 \leq \frac{D^2}{2\eta T}.$$

In particular,

$$\min_{t\in[T]}\left(\max_{\mathbf{y}\in\mathcal{K}} f(\mathbf{Z}_{t+1}, \mathbf{y}) - \min_{\mathbf{X}\in\mathcal{S}_n} f(\mathbf{X}, \mathbf{w}_{t+1})\right) \leq \frac{D^2}{2\eta T}.$$

Alternatively, using the convexity of $f(\cdot, \mathbf{y})$ and concavity of $f(\mathbf{X}, \cdot)$,

$$\max_{\mathbf{y}\in\mathcal{K}} f\left(\frac{1}{T}\sum_{t=1}^{T}\mathbf{Z}_{t+1}, \mathbf{y}\right) - \min_{\mathbf{X}\in\mathcal{S}_n} f\left(\mathbf{X}, \frac{1}{T}\sum_{t=1}^{T}\mathbf{w}_{t+1}\right)$$
$$\leq \max_{\mathbf{y}\in\mathcal{K}}\frac{1}{T}\sum_{t=1}^{T} f\left(\mathbf{Z}_{t+1}, \mathbf{y}\right) - \min_{\mathbf{X}\in\mathcal{S}_n}\frac{1}{T}\sum_{t=1}^{T} f\left(\mathbf{X}, \mathbf{w}_{t+1}\right)$$
$$\leq \frac{1}{T}\sum_{t=1}^{T}\max_{\mathbf{y}\in\mathcal{K}} f(\mathbf{Z}_{t+1}, \mathbf{y}) - \frac{1}{T}\sum_{t=1}^{T}\min_{\mathbf{X}\in\mathcal{S}_n} f(\mathbf{X}, \mathbf{w}_{t+1}) \leq \frac{D^2}{2\eta T}.$$

$\square$

### D.2 Proof of the main theorem — Theorem 1

To prove Theorem 1 we first prove two technical lemmas. We begin by proving that the iterates of Algorithm 1 always remain inside a ball of a certain radius around an optimal solution.

**Lemma 11.** *Let $\{(\mathbf{X}_t, \mathbf{y}_t)\}_{t\geq 1}$ and $\{(\mathbf{Z}_t, \mathbf{w}_t)\}_{t\geq 2}$ be the sequences generated by Algorithm 1 with a step-size $\eta_t \leq \frac{1}{\beta}$, and let $(\mathbf{X}^*, \mathbf{y}^*)$ be some optimal solution to Problem (4). Then for all $t \geq 1$ it holds that*

$$\|(\mathbf{X}_{t+1}, \mathbf{y}_{t+1}) - (\mathbf{X}^*, \mathbf{y}^*)\| \leq \|(\mathbf{X}_t, \mathbf{y}_t) - (\mathbf{X}^*, \mathbf{y}^*)\|,$$
$$\|(\mathbf{Z}_{t+1}, \mathbf{w}_{t+1}) - (\mathbf{X}^*, \mathbf{y}^*)\| \leq \left(1 + \frac{1}{\sqrt{1 - \eta_t^2\beta^2}}\right)\|(\mathbf{X}_t, \mathbf{y}_t) - (\mathbf{X}^*, \mathbf{y}^*)\|.$$

*Proof.* A known inequality of the EG algorithm (see for example Lemma 12.1.10 in [15]) is

$$\|(\mathbf{X}_{t+1}, \mathbf{y}_{t+1}) - (\mathbf{X}^*, \mathbf{y}^*)\|^2 \leq \|(\mathbf{X}_t, \mathbf{y}_t) - (\mathbf{X}^*, \mathbf{y}^*)\|^2 - (1 - \eta_t^2\beta^2)\|(\mathbf{X}_t, \mathbf{y}_t) - (\mathbf{Z}_{t+1}, \mathbf{w}_{t+1})\|^2.$$
$$(13)$$

Since $\eta_t^2 \beta^2 \leq 1$ it follows that

$$\|(\mathbf{X}_{t+1}, \mathbf{y}_{t+1}) - (\mathbf{X}^*, \mathbf{y}^*)\| \leq \|(\mathbf{X}_t, \mathbf{y}_t) - (\mathbf{X}^*, \mathbf{y}^*)\|.$$

In addition, using (13)

$$\|(\mathbf{X}_t, \mathbf{y}_t) - (\mathbf{Z}_{t+1}, \mathbf{w}_{t+1})\| \leq \sqrt{(1 - \eta_t^2 \beta^2)^{-1}} \|(\mathbf{X}_t, \mathbf{y}_t) - (\mathbf{X}^*, \mathbf{y}^*)\|.$$

Therefore,

$$
\begin{aligned}
\|(\mathbf{Z}_{t+1}, \mathbf{w}_{t+1}) - (\mathbf{X}^*, \mathbf{y}^*)\| &\leq \|(\mathbf{Z}_{t+1}, \mathbf{w}_{t+1}) - (\mathbf{X}_t, \mathbf{y}_t)\| + \|(\mathbf{X}_t, \mathbf{y}_t) - (\mathbf{X}^*, \mathbf{y}^*)\| \\
&\leq \sqrt{(1 - \eta_t^2 \beta^2)^{-1}} \|(\mathbf{X}_t, \mathbf{y}_t) - (\mathbf{X}^*, \mathbf{y}^*)\| + \|(\mathbf{X}_t, \mathbf{y}_t) - (\mathbf{X}^*, \mathbf{y}^*)\| \\
&= \left(1 + \frac{1}{\sqrt{1 - \eta_t^2 \beta^2}}\right) \|(\mathbf{X}_t, \mathbf{y}_t) - (\mathbf{X}^*, \mathbf{y}^*)\|.
\end{aligned}
$$

$\square$

We now prove that when close enough to a low-rank saddle-point of Problem (4), under an assumption of an eigen-gap in the gradient of the saddle-point, both projections onto the spectrahedron that are necessary in each iteration of Algorithm 1, result in low-rank matrices.

**Lemma 12.** *Let $(\mathbf{X}^*, \mathbf{y}^*)$ be an optimal solution to Problem (4). Let $\tilde{r}$ denote the multiplicity of $\lambda_n(\nabla_{\mathbf{X}} f(\mathbf{X}^*, \mathbf{y}^*))$ and for any $r \geq \tilde{r}$ denote $\delta(r) := \lambda_{n-r}(\nabla_{\mathbf{X}} f(\mathbf{X}^*, \mathbf{y}^*)) - \lambda_n(\nabla_{\mathbf{X}} f(\mathbf{X}^*, \mathbf{y}^*))$. Then, for any $\eta \geq 0$ and $(\mathbf{X}, \mathbf{y}) \in \mathcal{S}_n \times \mathcal{K}$, if*

$$
\begin{aligned}
&\|(\mathbf{X}, \mathbf{y}) - (\mathbf{X}^*, \mathbf{y}^*)\|_F \\
&\leq \frac{\eta}{1 + \sqrt{2}\eta \max\{\beta_X, \beta_{Xy}\} \left(1 + \frac{1}{\sqrt{1 - \eta^2 \beta^2}}\right)} \max\left\{\frac{\sqrt{\tilde{r}}\delta(r - \tilde{r} + 1)}{2}, \frac{\delta(r)}{(1 + 1/\sqrt{\tilde{r}})}\right\}
\end{aligned}
$$

*then rank $(\Pi_{\mathcal{S}_n}[\mathbf{X} - \eta\nabla_{\mathbf{X}} f(\mathbf{X}, \mathbf{y})]) \leq r$ and rank $(\Pi_{\mathcal{S}_n}[\mathbf{X} - \eta\nabla_{\mathbf{X}} f(\mathbf{Z}_+, \mathbf{w}_+)]) \leq r$ where $\mathbf{Z}_+ = \Pi_{\mathcal{S}_n}[\mathbf{X} - \eta\nabla_{\mathbf{X}} f(\mathbf{X}, \mathbf{y})]$ and $\mathbf{w}_+ = \Pi_{\mathcal{K}}[\mathbf{y} - \eta\nabla_{\mathbf{y}} f(\mathbf{X}, \mathbf{y})]$.*

*Proof.* Denote $\mathbf{P}^* = \mathbf{X}^* - \eta\nabla_{\mathbf{X}} f(\mathbf{X}^*, \mathbf{y}^*)$. By Lemma 6, $\nabla_{\mathbf{X}} f(\mathbf{X}^*, \mathbf{y}^*)$ is a subgradient of the corresponding nonsmooth objective $g(\mathbf{X}) = \max_{\mathbf{y} \in \mathcal{K}} f(\mathbf{X}, \mathbf{y})$ at the point $\mathbf{X}^*$. Moreover, this subgradient also satisfies the first-order optimality condition. Hence, invoking Lemma 3 with this subgradient we have that

$$
\begin{aligned}
&\forall i \leq \mathrm{rank}(\mathbf{X}^*) : \; \lambda_i(\mathbf{P}^*) = \lambda_i(\mathbf{X}^*) - \eta\lambda_n(\nabla_{\mathbf{X}} f(\mathbf{X}^*, \mathbf{y}^*)); \\
&\forall i > \mathrm{rank}(\mathbf{X}^*) : \; \lambda_i(\mathbf{P}^*) = -\eta\lambda_{n-i+1}(\nabla_{\mathbf{X}} f(\mathbf{X}^*, \mathbf{y}^*)). \quad (14)
\end{aligned}
$$

Therefore, using (14) and the fact that $\lambda_{n-i+1}(\nabla f_{\mathbf{X}}(\mathbf{X}^*, \mathbf{y}^*)) = \lambda_n(\nabla f_{\mathbf{X}}(\mathbf{X}^*, \mathbf{y}^*))$ for all $i \leq \tilde{r}$ we have,

$$
\begin{aligned}
\sum_{i=1}^{\tilde{r}} \lambda_i(\mathbf{P}^*) &= \sum_{i=1}^{\tilde{r}} \lambda_i(\mathbf{X}^* - \eta\nabla_{\mathbf{X}} f(\mathbf{X}^*, \mathbf{y}^*)) = \sum_{i=1}^{\tilde{r}} \lambda_i(\mathbf{X}^*) - \eta\sum_{i=1}^{\tilde{r}} \lambda_{n-i+1}(\nabla_{\mathbf{X}} f(\mathbf{X}^*, \mathbf{y}^*)) \\
&= \sum_{i=1}^{\mathrm{rank}(\mathbf{X}^*)} \lambda_i(\mathbf{X}^*) - \eta\sum_{i=1}^{\tilde{r}} \lambda_n(\nabla_{\mathbf{X}} f(\mathbf{X}^*, \mathbf{y}^*)) = 1 - \eta\tilde{r}\lambda_n(\nabla_{\mathbf{X}} f(\mathbf{X}^*, \mathbf{y}^*)). \quad (15)
\end{aligned}
$$

Let $\mathbf{P} \in \mathbb{S}^n$. From the structure of the Euclidean projection onto the spectrahedron (see Eq. (2)), it follows that a sufficient condition so that rank $(\Pi_{\mathcal{S}_n}[\mathbf{P}]) \leq r$ is that $\sum_{i=1}^{r} \lambda_i(\mathbf{P}) - r\lambda_{r+1}(\mathbf{P}) \geq 1$. We will bound the LHS of this inequality.

First, it holds that

$$\sum_{i=1}^{\tilde{r}} \lambda_i(\mathbf{P}) \underset{(a)}{\geq} \sum_{i=1}^{\tilde{r}} \lambda_i(\mathbf{P}^*) - \sum_{i=1}^{\tilde{r}} \lambda_i(\mathbf{P}^* - \mathbf{P}) \geq \sum_{i=1}^{\tilde{r}} \lambda_i(\mathbf{P}^*) - \sqrt{\tilde{r}\sum_{i=1}^{\tilde{r}} \lambda_i^2(\mathbf{P} - \mathbf{P}^*)}$$

$$\geq \sum_{i=1}^{\tilde{r}} \lambda_i(\mathbf{P}^*) - \sqrt{\tilde{r}\sum_{i=1}^{n} \lambda_i^2(\mathbf{P} - \mathbf{P}^*)} \geq \sum_{i=1}^{\tilde{r}} \lambda_i(\mathbf{P}^*) - \sqrt{\tilde{r}}\|\mathbf{P} - \mathbf{P}^*\|_F$$

$$\underset{(b)}{\geq} 1 - \eta\tilde{r}\lambda_n(\nabla_{\mathbf{x}}f(\mathbf{X}^*,\mathbf{y}^*)) - \sqrt{\tilde{r}}\|\mathbf{P} - \mathbf{P}^*\|_F, \tag{16}$$

where (a) holds from Ky Fan's inequality for eigenvalues and (b) holds from (15).

Now, for any $r \geq \tilde{r}$ using Weyl's inequality and (14)

$$\lambda_{r+1}(\mathbf{P}) \leq \lambda_{r+1}(\mathbf{P}^*) + \lambda_1(\mathbf{P} - \mathbf{P}^*) \leq \lambda_{r+1}(\mathbf{P}^*) + \|\mathbf{P} - \mathbf{P}^*\|_F$$
$$= -\eta\lambda_{n-r}(\nabla_{\mathbf{x}}f(\mathbf{X}^*,\mathbf{y}^*)) + \|\mathbf{P} - \mathbf{P}^*\|_F. \tag{17}$$

Thus, combining (16) and (17) we obtain

$$\sum_{i=1}^{r} \lambda_i(\mathbf{P}) - r\lambda_{r+1}(\mathbf{P}) \geq \sum_{i=1}^{\tilde{r}} \lambda_i(\mathbf{P}) - \tilde{r}\lambda_{r+1}(\mathbf{P}) \geq 1 + \eta\tilde{r}\delta(r) - (\tilde{r} + \sqrt{\tilde{r}})\|\mathbf{P} - \mathbf{P}^*\|_F. \tag{18}$$

Alternatively, if $r \geq 2\tilde{r} - 1$ then using the general Weyl inequality and (14) we obtain

$$\lambda_{r+1}(\mathbf{P}) \leq \lambda_{r-\tilde{r}+2}(\mathbf{P}^*) + \lambda_{\tilde{r}}(\mathbf{P} - \mathbf{P}^*) = \lambda_{r-\tilde{r}+2}(\mathbf{P}^*) + \sqrt{\lambda_{\tilde{r}}^2(\mathbf{P} - \mathbf{P}^*)}$$
$$\leq \lambda_{r-\tilde{r}+2}(\mathbf{P}^*) + \frac{1}{\sqrt{\tilde{r}}}\|\mathbf{P} - \mathbf{P}^*\|_F = -\eta\lambda_{n-r+\tilde{r}-1}(\nabla_{\mathbf{x}}f(\mathbf{X}^*,\mathbf{y}^*)) + \frac{1}{\sqrt{\tilde{r}}}\|\mathbf{P} - \mathbf{P}^*\|_F. \tag{19}$$

Thus, combining (16) and (19) we obtain

$$\sum_{i=1}^{r} \lambda_i(\mathbf{P}) - r\lambda_{r+1}(\mathbf{P}) \geq \sum_{i=1}^{\tilde{r}} \lambda_i(\mathbf{P}) - \tilde{r}\lambda_{r+1}(\mathbf{P}) \geq 1 + \eta\tilde{r}\delta(r - \tilde{r} + 1) - 2\sqrt{\tilde{r}}\|\mathbf{P} - \mathbf{P}^*\|_F. \tag{20}$$

Now we are left with bounding $\|\mathbf{P} - \mathbf{P}^*\|_F$. Note that by the smoothness of $f$, for any $(\mathbf{X},\mathbf{y}) \in \mathcal{S}_n \times \mathcal{K}$ it holds that

$$\|\nabla_{\mathbf{x}}f(\mathbf{X},\mathbf{y}) - \nabla_{\mathbf{x}}f(\mathbf{X}^*,\mathbf{y}^*)\|_F$$
$$\leq \|\nabla_{\mathbf{x}}f(\mathbf{X},\mathbf{y}) - \nabla_{\mathbf{x}}f(\mathbf{X}^*,\mathbf{y})\|_F + \|\nabla_{\mathbf{x}}f(\mathbf{X}^*,\mathbf{y}) - \nabla_{\mathbf{x}}f(\mathbf{X}^*,\mathbf{y}^*)\|_F$$
$$\leq \beta_X\|\mathbf{X} - \mathbf{X}^*\|_F + \beta_{Xy}\|\mathbf{y} - \mathbf{y}^*\|_2. \tag{21}$$

Taking $\mathbf{P} = \mathbf{X} - \eta\nabla_{\mathbf{x}}f(\mathbf{X},\mathbf{y})$ we get

$$\|\mathbf{P} - \mathbf{P}^*\|_F = \|\mathbf{X} - \eta\nabla_{\mathbf{x}}f(\mathbf{X},\mathbf{y}) - \mathbf{X}^* + \eta\nabla_{\mathbf{x}}f(\mathbf{X}^*,\mathbf{y}^*)\|_F$$
$$\leq \|\mathbf{X} - \mathbf{X}^*\|_F + \eta\|\nabla_{\mathbf{x}}f(\mathbf{X},\mathbf{y}) - \nabla_{\mathbf{x}}f(\mathbf{X}^*,\mathbf{y}^*)\|_F$$
$$\leq \|(\mathbf{X},\mathbf{y}) - (\mathbf{X}^*,\mathbf{y}^*)\| + \eta\|\nabla_{\mathbf{x}}f(\mathbf{X},\mathbf{y}) - \nabla_{\mathbf{x}}f(\mathbf{X}^*,\mathbf{y}^*)\|_F$$
$$\leq \|(\mathbf{X},\mathbf{y}) - (\mathbf{X}^*,\mathbf{y}^*)\| + \eta\beta_X\|\mathbf{X} - \mathbf{X}^*\|_F + \eta\beta_{Xy}\|\mathbf{y} - \mathbf{y}^*\|_2,$$

where the last inequality holds from (21).

For any $a, b \geq 0$ it holds that

$$a\|\mathbf{X} - \mathbf{X}^*\|_F + b\|\mathbf{y} - \mathbf{y}^*\|_2 \leq \max\{a,b\}\left(\|\mathbf{X} - \mathbf{X}^*\|_F + \|\mathbf{y} - \mathbf{y}^*\|_2\right)$$
$$\leq \sqrt{2}\max\{a,b\}\|(\mathbf{X},\mathbf{y}) - (\mathbf{X}^*,\mathbf{y}^*)\|.$$

Thus, by taking $a = \eta\beta_X$ and $b = \eta\beta_{Xy}$ we obtain

$$\|\mathbf{P} - \mathbf{P}^*\|_F \leq \left(1 + \sqrt{2}\eta\max\{\beta_X, \beta_{Xy}\}\right)\|(\mathbf{X}, \mathbf{y}) - (\mathbf{X}^*, \mathbf{y}^*)\|. \tag{22}$$

Therefore, plugging (22) into (18) we obtain that the condition $\sum_{i=1}^r \lambda_i(\mathbf{P}) - r\lambda_{r+1}(\mathbf{P}) \geq 1$ holds if

$$\|(\mathbf{X}, \mathbf{y}) - (\mathbf{X}^*, \mathbf{y}^*)\| \leq \frac{\eta\delta(r)}{\left(1 + 1/\sqrt{\tilde{r}}\right)\left(1 + \sqrt{2}\eta\max\{\beta_X, \beta_{Xy}\}\right)}.$$

Alternatively, plugging (22) into (20) we obtain that if $r \geq 2\tilde{r} - 1$ then the condition $\sum_{i=1}^r \lambda_i(\mathbf{P}) - r\lambda_{r+1}(\mathbf{P}) \geq 1$ holds if

$$\|(\mathbf{X}, \mathbf{y}) - (\mathbf{X}^*, \mathbf{y}^*)\| \leq \frac{\eta\sqrt{\tilde{r}}\delta(r - \tilde{r} + 1)}{2\left(1 + \sqrt{2}\eta\max\{\beta_X, \beta_{Xy}\}\right)}.$$

Note that $\delta(r - \tilde{r} + 1) > 0$ only if $r \geq 2\tilde{r} - 1$. Therefore, we can combine the last two inequalities to conclude that for any $r \geq \tilde{r}$ if

$$\|(\mathbf{X}, \mathbf{y}) - (\mathbf{X}^*, \mathbf{y}^*)\| \leq \frac{\eta}{1 + \sqrt{2}\eta\max\{\beta_X, \beta_{Xy}\}}\max\left\{\frac{\sqrt{\tilde{r}}\delta(r - \tilde{r} + 1)}{2}, \frac{\delta(r)}{(1 + 1/\sqrt{\tilde{r}})}\right\} \tag{23}$$

then $\text{rank}(\Pi_{\mathcal{S}_n}[\mathbf{X} - \nabla_{\mathbf{X}}f(\mathbf{X}, \mathbf{y})]) \leq r$.

Similarly, taking $\mathbf{P} = \mathbf{X} - \eta\nabla_{\mathbf{X}}f(\mathbf{Z}_+, \mathbf{w}_+)$ we get

$$\begin{aligned}
\|\mathbf{P} - \mathbf{P}^*\|_F &= \|\mathbf{X} - \eta\nabla_{\mathbf{X}}f(\mathbf{Z}_+, \mathbf{w}_+) - \mathbf{X}^* + \eta\nabla_{\mathbf{X}}f(\mathbf{X}^*, \mathbf{y}^*)\|_F \\
&\leq \|\mathbf{X} - \mathbf{X}^*\|_F + \eta\|\nabla_{\mathbf{X}}f(\mathbf{Z}_+, \mathbf{w}_+) - \nabla_{\mathbf{X}}f(\mathbf{X}^*, \mathbf{y}^*)\|_F \\
&\leq \|\mathbf{X} - \mathbf{X}^*\|_F + \eta\beta_X\|\mathbf{Z}_+ - \mathbf{X}^*\|_F + \eta\beta_{Xy}\|\mathbf{w}_+ - \mathbf{y}^*\|_2,
\end{aligned}$$

where the last inequality holds from (21).

For any $a, b, c \geq 0$ it holds that

$$\begin{aligned}
& a\|\mathbf{X} - \mathbf{X}^*\|_F + b\|\mathbf{Z}_+ - \mathbf{X}^*\|_F + c\|\mathbf{w}_+ - \mathbf{y}^*\|_2 \\
&\leq a\|\mathbf{X} - \mathbf{X}^*\|_F + \max\{b, c\}\left(\|\mathbf{Z}_+ - \mathbf{X}^*\|_F + \|\mathbf{w}_+ - \mathbf{y}^*\|_2\right) \\
&\leq a\|\mathbf{X} - \mathbf{X}^*\|_F + \sqrt{2}\max\{b, c\}\|(\mathbf{Z}_+, \mathbf{w}_+) - (\mathbf{X}^*, \mathbf{y}^*)\| \\
&\leq a\|(\mathbf{X}, \mathbf{y}) - (\mathbf{X}^*, \mathbf{y}^*)\| + \sqrt{2}\max\{b, c\}\|(\mathbf{Z}_+, \mathbf{w}_+) - (\mathbf{X}^*, \mathbf{y}^*)\| \\
&\leq \left(a + \sqrt{2}\max\{b, c\}\left(1 + \frac{1}{\sqrt{1 - \eta^2\beta^2}}\right)\right)\|(\mathbf{X}, \mathbf{y}) - (\mathbf{X}^*, \mathbf{y}^*)\|,
\end{aligned}$$

where the second to last inequality holds from Lemma 11.

Thus, by taking $a = 1$, $b = \eta\beta_X$, and $c = \eta\beta_{Xy}$ we obtain

$$\|\mathbf{P} - \mathbf{P}^*\|_F \leq \left(1 + \sqrt{2}\eta\max\{\beta_X, \beta_{Xy}\}\left(1 + \frac{1}{\sqrt{1 - \eta^2\beta^2}}\right)\right)\|(\mathbf{X}, \mathbf{y}) - (\mathbf{X}^*, \mathbf{y}^*)\|. \tag{24}$$

Therefore, plugging (24) into (18) we obtain that the condition $\sum_{i=1}^r \lambda_i(\mathbf{P}) - r\lambda_{r+1}(\mathbf{P}) \geq 1$ holds if

$$\|(\mathbf{X}, \mathbf{y}) - (\mathbf{X}^*, \mathbf{y}^*)\| \leq \frac{\eta\delta(r)}{\left(1 + 1/\sqrt{r^*}\right)\left(1 + \sqrt{2}\eta\max\{\beta_X, \beta_{Xy}\}\left(1 + \frac{1}{\sqrt{1-\eta^2\beta^2}}\right)\right)}.$$

Alternatively, plugging (24) into (20) we obtain that if $r \geq 2\tilde{r} - 1$ then the condition $\sum_{i=1}^r \lambda_i(\mathbf{P}) - r\lambda_{r+1}(\mathbf{P}) \geq 1$ holds if

$$\|(\mathbf{X}, \mathbf{y}) - (\mathbf{X}^*, \mathbf{y}^*)\| \leq \frac{\eta\sqrt{\tilde{r}}\delta(r - \tilde{r} + 1)}{1 + \sqrt{2}\eta\max\{\beta_X, \beta_{Xy}\}\left(1 + \frac{1}{\sqrt{1-\eta^2\beta^2}}\right)}.$$

Note that $\delta(r - \tilde{r} + 1) > 0$ only if $r \geq 2\tilde{r} - 1$. Therefore, we can combine the last two inequalities to conclude that for any $r \geq \tilde{r}$ if

$$\|(\mathbf{X}, \mathbf{y}) - (\mathbf{X}^*, \mathbf{y}^*)\|$$

$$\leq \frac{\eta}{1 + \sqrt{2}\eta \max\{\beta_X, \beta_{Xy}\} \left(1 + \frac{1}{\sqrt{1 - \eta^2\beta^2}}\right)} \max\left\{\frac{\sqrt{\tilde{r}}\delta(r - \tilde{r} + 1)}{2}, \frac{\delta(r)}{(1 + 1/\sqrt{\tilde{r}})}\right\} \quad (25)$$

then $\text{rank}(\Pi_{\mathcal{S}_n}[\mathbf{X} - \nabla_{\mathbf{X}} f(\mathbf{Z}_+, \mathbf{w}_+)]) \leq r$.

Taking the minimum between (23) and (25) gives us the bound on the radius in the lemma.

$\square$

### D.3 Proof of Theorem 1

*Proof of Theorem 1.* We will prove by induction that for all $t \geq 1$ it holds that $\|(\mathbf{X}_t, \mathbf{y}_t) - (\mathbf{X}^*, \mathbf{y}^*)\| \leq R_0(r)$ and $\|(\mathbf{Z}_t, \mathbf{w}_t) - (\mathbf{X}^*, \mathbf{y}^*)\| \leq (1 + \sqrt{2}) R_0(r)$, thus implying through Lemma 12 that all projections $\Pi_{\mathcal{S}_n}[\mathbf{X}_t - \eta\nabla_{\mathbf{X}} f(\mathbf{X}_t, \mathbf{y}_t)]$ and $\Pi_{\mathcal{S}_n}[\mathbf{X}_t - \eta\nabla_{\mathbf{X}} f(\mathbf{Z}_{t+1}, \mathbf{w}_{t+1})]$ can be replaced with their rank-r truncated counterparts given in (3), without any change to the result.

The initialization $\|(\mathbf{X}_1, \mathbf{y}_1) - (\mathbf{X}^*, \mathbf{y}^*)\| \leq R_0(r)$ holds trivially. Now, by Lemma 11, using recursion, we have that for all $t \geq 1$,

$$\|(\mathbf{X}_{t+1}, \mathbf{y}_{t+1}) - (\mathbf{X}^*, \mathbf{y}^*)\| \leq \|(\mathbf{X}_t, \mathbf{y}_t) - (\mathbf{X}^*, \mathbf{y}^*)\| \leq \cdots \leq \|(\mathbf{X}_1, \mathbf{y}_1) - (\mathbf{X}^*, \mathbf{y}^*)\| \leq R_0(r),$$

and for $\beta = \sqrt{2} \max\left\{\sqrt{\beta_X^2 + \beta_{yX}^2}, \sqrt{\beta_y^2 + \beta_{Xy}^2}\right\}$ we have that,

$$\|(\mathbf{Z}_{t+1}, \mathbf{w}_{t+1}) - (\mathbf{X}^*, \mathbf{y}^*)\| \leq \left(1 + \frac{1}{\sqrt{1 - \eta_t^2\beta^2}}\right)\|(\mathbf{X}_t, \mathbf{y}_t) - (\mathbf{X}^*, \mathbf{y}^*)\|$$

$$\leq \left(1 + \frac{1}{\sqrt{1 - \eta_t^2\beta^2}}\right)\|(\mathbf{X}_1, \mathbf{y}_1) - (\mathbf{X}^*, \mathbf{y}^*)\|$$

$$\leq (1 + \sqrt{2})\|(\mathbf{X}_1, \mathbf{y}_1) - (\mathbf{X}^*, \mathbf{y}^*)\| \leq (1 + \sqrt{2})R_0(r).$$

Therefore, under the assumptions of the theorem, Algorithm 1 can be run using only rank-r truncated projections, while maintaining its original convergence rate stated in Lemma 10. $\square$

## E Calculating the dual-gap in saddle-point problems

Set some point $(\widehat{\mathbf{Z}}, \widehat{\mathbf{w}}) \in \{\text{Tr}(\mathbf{X}) = \tau, \mathbf{X} \succeq 0\} \times \mathcal{K}$. Using the concavity of $f(\widehat{\mathbf{Z}}, \cdot)$ and convexity of $f(\cdot, \widehat{\mathbf{w}})$, for all $\mathbf{y} \in \mathcal{K}$ and $\mathbf{X} \in \{\text{Tr}(\mathbf{X}) = \tau, \mathbf{X} \succeq 0\}$, it holds that

$$f(\widehat{\mathbf{Z}}, \mathbf{y}) - f(\widehat{\mathbf{Z}}, \widehat{\mathbf{w}}) \leq \langle \widehat{\mathbf{w}} - \mathbf{y}, -\nabla_{\mathbf{y}} f(\widehat{\mathbf{Z}}, \widehat{\mathbf{w}})\rangle,$$

$$f(\widehat{\mathbf{Z}}, \widehat{\mathbf{w}}) - f(\mathbf{X}, \widehat{\mathbf{w}}) \leq \langle \widehat{\mathbf{Z}} - \mathbf{X}, \nabla_{\mathbf{X}} f(\widehat{\mathbf{Z}}, \widehat{\mathbf{w}})\rangle.$$

By taking the maximum of all $\mathbf{y} \in \mathcal{K}$ we obtain in particular that

$$f(\mathbf{X}^*, \mathbf{y}^*) - f(\widehat{\mathbf{Z}}, \widehat{\mathbf{w}}) \leq f(\widehat{\mathbf{Z}}, \mathbf{y}^*) - f(\widehat{\mathbf{Z}}, \widehat{\mathbf{w}}) \leq \max_{\mathbf{y} \in \mathcal{K}} f(\widehat{\mathbf{Z}}, \mathbf{y}) - f(\widehat{\mathbf{Z}}, \widehat{\mathbf{w}})$$

$$\leq \max_{\mathbf{y} \in \mathcal{K}} \langle \widehat{\mathbf{w}} - \mathbf{y}, -\nabla_{\mathbf{y}} f(\widehat{\mathbf{Z}}, \widehat{\mathbf{w}})\rangle,$$

and taking the maximum of all $\mathbf{X} \in \{\text{Tr}(\mathbf{X}) = \tau, \mathbf{X} \succeq 0\}$

$$f(\widehat{\mathbf{Z}}, \widehat{\mathbf{w}}) - f(\mathbf{X}^*, \mathbf{y}^*) \leq f(\widehat{\mathbf{Z}}, \widehat{\mathbf{w}}) - f(\mathbf{X}^*, \widehat{\mathbf{w}}) \leq f(\widehat{\mathbf{Z}}, \widehat{\mathbf{w}}) - \min_{\substack{\text{Tr}(\mathbf{X}) = \tau, \\ \mathbf{X} \succeq 0}} f(\mathbf{X}, \widehat{\mathbf{w}})$$

$$\leq \max_{\substack{\text{Tr}(\mathbf{X}) = \tau, \\ \mathbf{X} \succeq 0}} \langle \widehat{\mathbf{Z}} - \mathbf{X}, \nabla_{\mathbf{X}} f(\widehat{\mathbf{Z}}, \widehat{\mathbf{w}})\rangle.$$

Summing these two inequalities, we obtain a bound on the dual-gap at $(\widehat{\mathbf{Z}}, \widehat{\mathbf{w}})$ which can be written as

$$g(\widehat{\mathbf{Z}}) - g^* \leq \max_{\mathbf{y} \in \mathcal{K}} f(\widehat{\mathbf{Z}}, \mathbf{y}) - \min_{\mathbf{X} \in \mathcal{S}_n} f(\mathbf{X}, \widehat{\mathbf{w}})$$
$$\leq \max_{\substack{\mathrm{Tr}(\mathbf{X}) = \tau, \\ \mathbf{X} \succeq 0}} \langle \widehat{\mathbf{Z}} - \mathbf{X}, \nabla_{\mathbf{X}} f(\widehat{\mathbf{Z}}, \widehat{\mathbf{w}}) \rangle - \min_{\mathbf{y} \in \mathcal{K}} \langle \widehat{\mathbf{w}} - \mathbf{y}, \nabla_{\mathbf{y}} f(\widehat{\mathbf{Z}}, \widehat{\mathbf{w}}) \rangle.$$

It is easy to see that the maximizer of the first term in the RHS of the above is $\tau \mathbf{v}_n \mathbf{v}_n^\top$ where $\mathbf{v}_n$ is the smallest eigenvector of $\nabla_{\mathbf{X}} f(\widehat{\mathbf{Z}}, \widehat{\mathbf{W}})$, and the minimizer of the second term is $\mathbf{Y}_{i,j} = \mathrm{sign}(\nabla_{\mathbf{Y}} f(\widehat{\mathbf{Z}}, \widehat{\mathbf{W}})_{i,j})$ for $\mathcal{K} = \{\mathbf{Y} \in \mathbb{R}^{n \times n} \mid \|\mathbf{Y}\|_\infty \leq 1\}$ and $\nabla_{\mathbf{y}} f(\widehat{\mathbf{Z}}, \widehat{\mathbf{w}}) / \|\nabla_{\mathbf{y}} f(\widehat{\mathbf{Z}}, \widehat{\mathbf{w}})\|_2$ for $\mathcal{K} = \{\mathbf{y} \in \mathbb{R}^n \mid \|\mathbf{y}\|_2 \leq 1\}$.

# F   Additional empirical evidence

## F.1   Additional numerical results for sparse PCA

See Table 5.

Table 5: Additional numerical results for the sparse PCA problem.

| dimension (n) | 100 | 200 | 400 | 600 |
|---|---|---|---|---|
| $\downarrow \mathbf{N}_{ij} \sim U[0,1]$, SNR = 1 $\downarrow$ | | | | |
| $\lambda$ | 0.008 | 0.004 | 0.002 | 0.0013 |
| initialization error | 0.5997 | 0.6009 | 0.5990 | 0.6002 |
| recovery error | 0.0054 | 0.0040 | 0.0035 | 0.0043 |
| dual gap | $4.1 \times 10^{-5}$ | $7.9 \times 10^{-5}$ | $4.9 \times 10^{-5}$ | $3.4 \times 10^{-6}$ |
| gap($\nabla_{\mathbf{X}} f(\mathbf{X}^*, \mathbf{y}^*)$) | 0.8840 | 0.8898 | 0.8938 | 0.8777 |
| $\downarrow \mathbf{N}_{ij} \sim \mathcal{N}(0.5, \mathbf{I}_n)$, SNR = 1 $\downarrow$ | | | | |
| $\lambda$ | 0.006 | 0.003 | 0.0015 | 0.001 |
| initialization error | 0.1584 | 0.1464 | 0.1443 | 0.1411 |
| recovery error | 0.0059 | 0.0033 | 0.0019 | 0.0015 |
| dual gap | $8.6 \times 10^{-4}$ | 0.0031 | 0.0053 | 0.0060 |
| gap($\nabla_{\mathbf{X}} f(\mathbf{X}^*, \mathbf{y}^*)$) | 0.8406 | 0.8869 | 0.9178 | 0.9331 |
| $\downarrow \mathbf{N}_{ij} \sim \mathcal{N}(0.5, \mathbf{I}_n)$, SNR = 0.05 $\downarrow$ | | | | |
| $\lambda$ | 0.04 | 0.02 | 0.01 | 0.005 |
| initialization error | 1.6701 | 1.6620 | 1.6542 | 1.6610 |
| recovery error | 0.0502 | 0.0234 | 0.0137 | 0.0109 |
| dual gap | $1.9 \times 10^{-5}$ | 0.0041 | 0.0534 | 0.0409 |
| gap($\nabla_{\mathbf{X}} f(\mathbf{X}^*, \mathbf{y}^*)$) | 0.2200 | 0.4076 | 0.5460 | 0.6788 |

## F.2   Additional numerical results for low-rank and sparse matrix recovery

See Table 6.

## F.3   Additional numerical results for robust PCA

We test the model also with $\mathrm{rank}(\mathbf{Z}_0 \mathbf{Z}_0^\top) = 1, 5$. For $\mathrm{rank}(\mathbf{Z}_0 \mathbf{Z}_0^\top) = 1$ we set the step-size to $\eta = n/10$ and for $\mathrm{rank}(\mathbf{Z}_0 \mathbf{Z}_0^\top) = 5$ we set $\eta = 1$. For each set of parameters the measurements are averaged over 10 i.i.d. runs. See Table 7.

Table 6: Numerical results for the low-rank and sparse matrix recovery problem.

| dimension (n) | 100 | 200 | 400 | 600 |
|---|---|---|---|---|
| $\downarrow r = \mathrm{rank}(\mathbf{Z}_0\mathbf{Z}_0^\top) = 1$, SNR $= 0.48\downarrow$ | | | | |
| $\lambda$ | 0.0012 | 0.0035 | 0.0016 | 0.001 |
| initialization error | 0.4562 | 0.4471 | 0.4507 | 0.4450 |
| recovery error | 0.0364 | 0.0193 | 0.0160 | 0.0168 |
| dual gap | 0.0083 | 0.0086 | 0.0020 | $4.2 \times 10^{-4}$ |
| gap$(\nabla_{\mathbf{X}} f(\mathbf{X}^*, \mathbf{y}^*))$ | 0.0628 | 0.1439 | 0.1258 | 0.1069 |

Table 7: Numerical results for the robust PCA problem.

| dimension (n) | 100 | 200 | 400 | 600 |
|---|---|---|---|---|
| $\downarrow r = \mathrm{rank}(\mathbf{Z}_0\mathbf{Z}_0^\top) = 1, T = 3000\downarrow$ | | | | |
| SNR | 0.0021 | $7.2 \times 10^{-4}$ | $2.5 \times 10^{-4}$ | $1.3 \times 10^{-4}$ |
| initialization error | 1.3511 | 1.3430 | 1.2889 | 1.2606 |
| recovery error | 0.0084 | 0.0107 | 0.0109 | 0.0107 |
| dual gap | 0.0016 | 0.0029 | 0.0044 | 0.0069 |
| gap$(\nabla_{\mathbf{X}} f(\mathbf{X}^*, \mathbf{y}^*))$ | 15.5944 | 41.2139 | 85.8117 | 140.5349 |
| $\downarrow r = \mathrm{rank}(\mathbf{Z}_0\mathbf{Z}_0^\top) = 1, T = 3000\downarrow$ | | | | |
| SNR | 0.0110 | 0.0038 | 0.0013 | $6.9 \times 10^{-4}$ |
| initialization error | 1.5501 | 1.5527 | 1.5221 | 1.4833 |
| recovery error | 0.0092 | 0.0092 | 0.0087 | 0.0075 |
| dual gap | 0.0084 | 0.0390 | 0.1866 | 0.4721 |
| gap$(\nabla_{\mathbf{X}} f(\mathbf{X}^*, \mathbf{y}^*))$ | 7.6734 | 26.2132 | 66.1113 | 108.7215 |

## F.4 Phase synchronization

We consider the phase synchronization problem (see for instance [41]) which can be written as:

$$\max_{\substack{\mathbf{z} \in \mathbb{C}^n, \\ |z_j| = 1\ \forall j \in [n]}} \mathbf{z}^*\mathbf{M}\mathbf{z}, \tag{26}$$

where $\mathbf{M} = \mathbf{z}_0\mathbf{z}_0^* + c\mathbf{N}$ is a noisy observation of some rank-one matrix $\mathbf{z}_0\mathbf{z}_0^*$ such that $\mathbf{z}_0 \in \mathbb{C}^n$ and $\mathbf{z}_{0j} = e^{i\theta_j}$ where $\theta_j \in [0, 2\pi]$. We follow the statistical model in [41] where the noise matrix $\mathbf{N} \in \mathbb{C}^{n \times n}$ is chosen such that every entry is

$$\mathbf{N}_{jk} = \begin{cases} \mathcal{N}(0,1) + i\mathcal{N}(0,1) & j < k \\ \overline{\mathbf{N}}_{kj} & j > k \\ 0 & j = k \end{cases}.$$

It is known that for a large $n$ and $c = \mathcal{O}\left(\sqrt{\frac{n}{\log n}}\right)$, with high probability the SDP relaxation of (26) is able to recover the original signal (see [41]).

We solve a penalized version of the SDP relaxation of (26) which can be written as the following saddle-point optimization problem:

$$\min_{\substack{\mathrm{Tr}(\mathbf{X})=n, \\ \mathbf{X} \succeq 0}} \langle \mathbf{X}, -\mathbf{M} \rangle + \lambda \|\mathrm{diag}(\mathbf{X}) - \overrightarrow{\mathbf{1}}\|_2 = \min_{\substack{\mathrm{Tr}(\mathbf{X})=n, \\ \mathbf{X} \succeq 0}} \max_{\|\mathbf{y}\|_2 \leq 1} \langle \mathbf{X}, -\mathbf{M} \rangle + \lambda \langle \mathrm{diag}(\mathbf{X}) - \overrightarrow{\mathbf{1}}, \mathbf{y} \rangle,$$

where $\overrightarrow{\mathbf{1}}$ is the all-ones vector.

While the phase synchronization problem is formulated over the complex numbers, extending our results to handle this model is straightforward.

We initialize the $\mathbf{X}$ variable with the rank-one approximation of $\mathbf{M}$. That is, we take $\mathbf{X}_1 = n\mathbf{u}_1\mathbf{u}_1^*$, where $\mathbf{u}_1$ is the top eigenvector of $\mathbf{M}$. For the $\mathbf{y}$ variable we initialize it with $\mathbf{y}_1 = (\text{diag}(\mathbf{X}_1) - \overrightarrow{\mathbf{1}})/\|\text{diag}(\mathbf{X}_1) - \overrightarrow{\mathbf{1}}\|_2$.

We set the noise level to $c = 0.18\sqrt{n}$. We set the number of iterations in each experiment to $T = 10,000$ and for each choice of $n$ we average the measurements over 10 i.i.d. runs.

Table 8: Numerical results for the phase synchronization problem.

| dimension (n) | 100 | 200 | 400 | 600 |
|---|---|---|---|---|
| SNR | 0.1553 | 0.0775 | 0.0387 | 0.0258 |
| $\lambda$ | 200 | 600 | 1600 | 2800 |
| $\eta$ | 1/400 | 1/800 | 1/1800 | 1/1800 |
| initialization error | 0.1270 | 0.1255 | 0.1284 | 0.1323 |
| recovery error | 0.0698 | 0.0659 | 0.0683 | 0.0719 |
| dual gap | $7.8 \times 10^{-8}$ | $3.9 \times 10^{-5}$ | 0.1553 | 0.5112 |
| gap($\nabla_{\mathbf{X}} f(\mathbf{X}^*, \mathbf{y}^*)$) | 39.8591 | 78.9982 | 150.3524 | 217.06 |
| $\|\text{diag}(\mathbf{X}^*) - \overrightarrow{\mathbf{1}}\|_2$ | $3.2 \times 10^{-10}$ | $2.1 \times 10^{-8}$ | $5.1 \times 10^{-7}$ | $3.7 \times 10^{-7}$ |