# OpenReview forum: "Low-Rank Extragradient Method for Nonsmooth and Low-Rank Matrix Optimization Problems"
_NeurIPS.cc/2021/Conference — NeurIPS 2021 Poster_

### Official Review · Reviewer_u8Jo · 2021-07-14

**Rating:** 5
**Confidence:** 2

**Summary:**

This paper develops an extragradient method for low-rank and nonsmooth matrix optimization problems. Under certain conditions, the proposed method converges to an optimal solution with rate O(1/t). Numerical experiments are provided to support the theoretical results.

**Limitations And Societal Impact:**

Yes

**Main Review:**

1. I think authors should use more space to the assumptions imposed on the objective function and the optimal solution. The current version is not straightforward to see if the assumptions are stringent or easy to hold. For example, it would make paper more motived to show some applications in the introduction where the assumptions do hold.
2. It would be better to discuss the initialization condition in Theorem 1. Is this initialization requirement easy to meet?
3. In the numerical experiment, there is no comparison with the state-of-the-art algorithms. Therefore, it is unknown if the proposed algorithm improves over the prior work. As far as I know, there exist more efficient algorithms (converges faster than O(1/t)) to solve the applications in the numerical experiment.

As a result, I tend to vote for rejection.

**Time Spent Reviewing:**

4

---

> ### Author Response · Authors · 2021-08-09
> **reply**
>
> Thank you for your comments and time invested in reviewing our work.
>
> 1.	Regarding our assumptions, please see Item 1 in our response to Reviewer MxNL and Item 1 in our response to Reviewer xRc5. Please also note that in Section 5 (and the corresponding part of the appendix) we bring several detailed applications.
>
> 2.	Regarding initialization, as we show in all the applications in Section 5, using a simple initialization scheme worked very well for all of our experiments (five tasks with different noise models and dimension). Nevertheless, since our theoretical results hinges on generalized strict complementarity, if the requirement does not hold for r^* exactly, it is possible to slightly increase the rank of the SVD computations which can significantly increase the required radius in the theorem (see expression for R_0(r) in Theorem 1). Please see also Item 3 in our response to Reviewer xRc5.
>
> 3.	To the best of our knowledge there are no existing methods that guarantee a faster rate than O(1/t) for the nonsmooth optimization problems under consideration in this paper, at least without additional strong assumptions on the problem, such as strong convexity. In particular, for the class of problems under consideration in this work, in terms of convergence rate, the extragradient method is state-of-the-art. In fact, we are not aware of other first-order methods for the general class of problems under consideration that do not require storing or factorizing high-rank matrices, even under our assumptions.
> The novelty of our work is that we show that, under a generalized strict complementarity assumption, all SVD computations that are necessary when computing the step of the known state-of-the-art extragradient method can be replaced with low-rank SVD computations, and the iterates that will be obtained throughout the run will be the same as would have been if using full SVDs. Therefore, this was our main concern throughout the experiments in section 5.

---

### Official Review · Reviewer_zvmw · 2021-07-16

**Rating:** 6
**Confidence:** 5

**Summary:**

This paper develops an algorithm for nonsmooth convex objectives with a trace norm ball constraint. It produces an algorithm using the extragradient method and shows the method enjoys lower iteration complexity due to strict complementarity.

**Limitations And Societal Impact:**

I did not see the author mentioned any limitation of the current work.

**Main Review:**

I think this is a good paper and it solves a class of nonsmooth problems that are interesting. I have a few questions that would like to hear the author's comment on them.

1. Theoretically, the algorithm needs a warm start. Even though I think warm starting the matrix variable X is easy. It is not clear to me how to initialize the vector variable y. Can the author comment on why the y is initialized in the way in Section 5? Is there any rule being used for initializing y in section 5?

2. Is there any particular reason to use the extragradient method?  Is it because simpler methods that use only one matrix variable and one vector variable, and one gradient update in each iteration, cannot have the same convergence guarantees as the extragradient method in general?

3. Does the 1/t rate appears in the examples in section 5? Or do you observe an even faster convergence?

**Time Spent Reviewing:**

2

---

> ### Author Response · Authors · 2021-08-09
> **reply**
>
> Thank you for your comments and time invested in reviewing our work.
>
> 1.	We do not have a theoretical proof of how best to initialize the y variable. The heuristic we used in the applications in section 5 is to take the y that maximizes the nonsmooth term for the initial X we chose. For example, if the nonsmooth term was the \ell_1-norm, we took y_1=sign(X_1) since it exactly satisfies
> \max_{||y||_{inf}<=1} <y,X_1>  = <y_1,X_1>=||X_1||_1.
> In practice, for all applications we present in section 5 this heuristic worked very well and the certificate that guarantees that the projections will return a low-rank matrix held starting from the initialization itself.
>
> 2.	To the best of our knowledge there are no methods that obtain a O(1/t) convergence rate for nonsmooth problems that do not take advantage of the ability to reformulate the problem as a saddle-point problem or have additional assumptions. Even in the class of methods that operate on the saddle-point formulation, to the best of our knowledge, there is no method that performs a single primal step and a single dual step and obtains the O(1/t) rate (obtaining the slower O(1/sqrt(t)) rate is possible via a single primal and dual update).
>
> 3.	The O(1/t) convergence rate of the extragradient method is a well-known result that has been discussed in many papers and is not our novel contribution. Therefore, we did not feel it is necessary to demonstrate its convergence rate in our experiments. The experiments mostly concern the truly novel ingredient component of our work, that under a generalized strict complementarity assumption, all SVD computations that are necessary when computing the known extragradient method can be replaced with low-rank SVD computations, and the iterates that will be computed throughout the run will be the same as they would have been if using full SVDs. i.e., completely identical to a standard naïve implementation of the extragradient method.

---

> > ### Comment · Reviewer_zvmw · 2021-08-31
> > **clarification of point 3**
> >
> > Hi Authors,
> >
> > I understand that the novel part is not the convergence rate. So the third question is more of curiosity instead of saying you should include this experiment. Can you tell me whether you observe faster convergence in practice?

---

> > > ### Author Response · Authors · 2021-09-01
> > > **reply**
> > >
> > > We did not check the convergence rates at all, so we can't say for sure.

---

### Official Review · Reviewer_MxNL · 2021-07-17

**Rating:** 5
**Confidence:** 3

**Summary:**

This paper considers low-rank and nonsmooth optimization problems and proposes the extragradient method to solve it.

**Limitations And Societal Impact:**

This is a theoretical paper, and I think there is no potential negative societal impact.

**Main Review:**

Strengths:

The extragradient method is proved to converge with rate $O(1/t)$ and low computational cost.

Weakness:

1. There seems no discussion on how restrictive the generalized strict complementarity condition (Assumption 1) in the paper. It would be much better if more explanations and intuitions about this condition are given.

2.  It's said that an advantage of the extragradient method over previous standard methods is low per iteration computational cost. How faster is this method than others quantitatively?

**Time Spent Reviewing:**

1.5

---

> ### Author Response · Authors · 2021-08-09
> **reply**
>
> Thank you for your comments and time invested in reviewing our work.
>
> 1.	Strict complementarity is a well-studied concept in continuous optimization theory which concerns the celebrated KKT conditions and is present in many works (see for instance the following influential works that : Zirui Zhou and Anthony Man-Cho So “A Unified Approach to Error Bounds for Structured Convex Optimization Problems”, 2015 and Dmitriy Drusvyatskiy and Adrian S. Lewis “Error bounds, quadratic growth, and linear convergence of proximal methods”, 2016).
> In our setting, of convex minimization over the Spectrahedron, it is shown that strict complementarity almost always holds (see for instance lemma 8 in Lijun Ding, Yingjie Fei, Qiantong Xu, and Chengrun Yang “Spectral Frank-Wolfe Algorithm: Strict Complementarity and Linear Convergence”, 2020), and therefore, even a standard strict complementarity assumption would be a reasonable assumption to make. Yet, our generalized strict complementarity assumption is much weaker. We merely assume the existence of a gap in the eigenvalues of the gradient of the optimal solution between the smallest and r-th smallest eigenvalues for some r>=r^*. Since X^* is a rank-r^* optimal solution, even if there is not a significant enough gap between the smallest and r^*-th smallest eigenvalues of the gradient of the optimal solution, it is very likely that slightly increasing r will enlarge the gap by a significant amount.
> There is a tradeoff between the rank of the SVD computations and the radius of the ball around the optimal solution inside-which the low-rank SVD computations can be used. This radius increases with the eigen-gap in the gradient of the optimal solution times \sqrt{r^*}. Thus, for a “warm-start” initialization to be contained in the relevant ball around the optimal solution, the rank of the SVD computations might need to be moderately increased. However, finding a relevant rank of the SVD computations should be possible, and hence, not including highly pathological and engineered cases, the generalized strict complementarity condition does not seem restrictive.
>
> 2.	The O(1/t) convergence rate of the extragradient method is a well-known result that has been discussed in many papers (in contrast to the subgradient method that only has a O(1/sqrt(t)) rate). It is well-known that it can be used to solve nonsmooth optimization problems that can be written as a smooth saddle-point problem. The novelty of our work is that we show that, under a generalized strict complementarity assumption, all SVD computations that are necessary when computing the known extragradient method can be replaced with low-rank SVD computations, and the iterates that will be obtained throughout the run will be the same as would have been obtained if using full SVDs. The low-rank SVD computations are what allow for the low per-iteration computational cost. The quantitative difference in run-time between computing full SVD computations (O(n^3)) and low-rank SVD computations (approximately O(rn^2)) is also well-known. We will discuss this more clearly in the revised version.

---

### Official Review · Reviewer_xRc5 · 2021-08-01

**Rating:** 5
**Confidence:** 4

**Summary:**

This paper considers the problem of minimizing a non-smooth matrix function  $f(X)$ over the spectrahedron $S_n$, i.e., positive semidefinite matrices with trace equal to 1. Such optimization problems often arise as the convex relaxation of various low-rank matrix recovery problems. In general solving these problems with projected subgradient descent requires a projection onto the spectraherdon at every iteration, which is expensive because of the need to compute a full SVD at every step.

Under the assumption that the non-smooth objective function $f(X)$ can be written as the pointwise maximum of a family of smooth functions,  the authors reformulate the original optimization problem as a saddle point problem which can be solved via the projected extragradient method. The key contribution is that under a “generalized strict complementarity” condition and when close to the ground truth, the projection onto $S_n$ can be replaced by a low-rank projection. Therefore, the overall algorithm can be significantly more efficient, because only a low-rank SVD is needed.

**Limitations And Societal Impact:**

Yes.

**Main Review:**

Pros:
1.	Prior work by Garber 2019 [16,17] consider only smooth functions, which has relatively limited practical applications for machine learning. This paper expands the scope by considering nonsmooth functions and gives examples like robust PCA and sparse PCA, which are of interest for the ML community.
2.	Gives an interesting counterexample where previous guarantee for projected gradient descent does indeed fail for projected subgradient descent. This justifies the use of the extragradient method proposed in this paper.
3.	The saddle-point view has further applications in the context of two-player zero sum games, and other minimax type problems in ML.
4.	This paper also establishes an interesting tradeoff between the rank of the SVD and the radius of the neighborhood around the ground truth in which the extragradient method converges.

Cons:
1. The setting of this paper is fairly restrictive. The main result of this paper only applies to functions of the form $g(X) = h(X)+\max_y y^T(\mathcal{A}(x)-b)$, where $h(X)$ is a smooth convex function and $\mathcal{A}$ is a linear map.

2. The proof techniques seem incremental in view of the previous work of [16] and [17]. It is not clear how these techniques can be further extended to more general non-smooth functions.

3. The authors do not investigate how a good initial point might be found that satisfy the assumptions of their theoretical results. Also it seems difficult to verify a priori whether the strict complementarity condition actually holds for a given problem, without already having solved it.

4. Moreover, the numerical experiments in section 5 do not demonstrate the advantage of the extragradient method. They do not show the convergence rate of the method. It would be nice if we can see a more straightforward comparison between the convergence rates and computational costs of the extragradient method and projected subgradient descent.

Minor comments:
1. I suggest that the authors define the generalized strict complementarity condition and the standard complementarity condition more clearly in the introduction. Right now both terms are used in the introduction without a clear definition, which is a bit confusing. There are numerous places where strict complementarity is referred to, and it is not clear which one is being talked about.
2. I also feel that section 2 needs a bit more motivation. It is not clear from this section why the strict complementarity condition is important and what Lemma 3 and 4 are used for.


----- Update after rebuttal -----
I have read the response of the authors. As before, my main concern is that the proposed algorithm does not have strong guarantees of success. Therefore I choose to keep my original score.

**Time Spent Reviewing:**

8

---

> ### Author Response · Authors · 2021-08-09
> **reply**
>
> Thank you for your comments and time invested in reviewing our work.
>
> 1.	As we show, there are many problems of interest that satisfy the nonsmooth structure considered in Assumption 2 and many problems in statistics / ML follow such structure, in particular when the nonsmooth part comes from a norm regularizer, and these are already very challenging to handle in the low-rank setting efficiently.
> Moreover, Assumption 2 is only necessary to translate the strict complementarity of the nonsmooth problem (i.e., Assumption 1) to strict complementarity of the saddle-point problem (i.e., Assumption 3). if we assume strict complementarity holds directly for the saddle-point formulation, then Assumption 2 is not necessary at all, and our results hold even without it. For instance, by working directly with Assumption 3 and a very standard Slater-point condition (i.e., the existence of a strictly feasible point), our method can tackle general functional constraints (either equalities or inequalities) by directly tackling the resulting Lagrangian  (see for example “Subgradient Methods for Saddle-Point Problems”, Angelia Nedi´c and Asuman Ozdaglar, 2007). We will add these discussions in detail to our revised version
>
> 2.	On the technical level , our work extends [16], [17], and although some of the analysis extends using similar arguments to the smooth case, there was work in the proof of the analysis for the extragradient method that is not straightforward from the analysis in [16] and [17]. For example, to the best of our knowledge we are the first to prove that both series of iterates generated in the extragradient method {(X_t,y_t)} and {(Z_t,w_t)} always remain inside the relevant ball around an optimal solution. This property is crucial for obtaining our main theorem. Our analysis is also complicated by the fact that we have two sequences and that unlilke [16],[17], the update to X_t is with the gradient w.r.t. to Z_t, etc.
> On a methodological level, perhaps our main contribution (as we see it), and in light of the negative result in Lemma 5 (regarding extension of projected subgradient), is to show that the “correct way” the address such low-rank nonsmooth problems is via the saddle-point formulation and we believe this is an important observation that our work makes with the appropriate supporting formalism.
>
> 3.	Indeed, we do not have a theoretical result on how to provide provably efficient initialization. We address this important issue in two ways: 1. We provide extensive numerical experiments on many tasks and considering different noise settings and dimension to demonstrate that empirically, it sees that very simple and natural initialization schemes seem to work well in practice. 2. More importantly, as we discuss in section 4.1, it is not necessary to know a priori whether the strict complementarity condition holds, since we have an easily computable certificate that can verify the correctness of the low-rank projection. This certificate could be easily checked on each iteration and thus enables to  verify the correct convergence of the algorithm in a practical manner, without worrying about either the strict complementarity or the initialization.
>
> 4.	The convergence rate of the extragradient method and its advantage over projected subgradient decent are well known results that have been established in many papers and text books. In particular, for the class of problems considered in this paper, in terms of convergence rate, the extragradient method is the state-of-the-art.  We consider Projected subgradient decent in our discussions mainly form a theoretical point of view, since it is the simplest and most general first-order method for nonsmooth problems, but it is well known that it is not very efficient in practice, in particular, for problems with favorable structure.
> For example, projected subgradient descent has convergence rate of O(1/\sqrt{T}) (even when problem admits smooth saddle-point formulation), while extragradient method (which takes advantage of the saddle-point structure) has a faster rate of O(1/T). Since these are well known results, we do not feel it is required to demonstrate the convergence rate of the extragradient method or compare it to projected subgradient decent. Moreover, as we demonstrate in the paper, even under our assumption it cannot be guaranteed that the projected subgradient method won't require storing and factorizing high-rank matrices, as opposed to the proposed extragradient method.
> We will make this point clearer in our discussion in the revised version.
> Our experiments mainly focus on showing that our theoretical results regarding the plausibility of using only low-rank SVDs indeed seem to also hold in practice on a variety of applications and settings.
>
> Minor comments:
>
> 1.	Thanks, we will add a definition of strict complementarity to the introduction and clarify the type of strict complementarity referred to throughout the paper.
> 2.	Our results depend strongly on the generalized strict complementarity condition. This assumption is necessary to ensure the existence of a radius around the optimal solution in which all SVD projections can be replaced with low-rank SVDs. Lemma 4 motivates the importance of the strict complementarity assumption: it demonstrates that problems that do not satisfy strict complementarity are ill-conditioned in the sense that they are not robust to noise in the parameters of the problem. Lemma 3 is a necessary characteristic of the gradient of the optimal solution that enables us to derive the bound on the radius around the optimal solution for which low-rank SVDs can be used, and it essential to the proof of our main theorem.

---

> > ### Author Response · Authors · 2021-09-05
> > **response after update of review**
> >
> > In his updated review, the reviewer concludes: "proposed algorithm does not have strong guarantees of success".
> > It is important to understand that we are attempting to tackle a very difficult and quite general low-rank optimization problems via highly efficient methods - such that do not require to maintain high rank matrices.
> > Most results in this area are tacking a statistical approach, that is, assuming a very specific model of the data and the loss function, and only then give efficient methods that most often converge from a warm-start, and some times, under suitable assumptions, do not require warm-start.
> > Thus, our result should be qualitatively compared to such ideas, and not black box methods for convex optimization, which have the mildest assumptions, but are completely impractical in large scale.
> > Differently from the above mentioned statistical approaches, we do not assume any generative mechanism on the data or a very specific loss functions (such as least squares), or even a smooth objective!
> > We only require the generalized strict complementarity assumptions which is motivated both theoretically and most importantly empirically, and we do have warm-start requirement has most of the research on efficient methods for low-rank problems with provable rate of convergence.

---

### Decision · Program_Chairs · 2021-09-28

**Decision:**

Accept (Poster)

**Comment:**

Thank you for your submission to NeurIPS. Overall, all reviewers found the work interesting and valuable, but could not agree on whether it was ready for publication. This was a difficult decision -- while the paper provides an interesting theoretical analysis, the reviewers found the setting to be quite restrictive, and the exposition of the assumptions to be unclear. One reviewer who reviewed the proof in detail expressed some concern in the amount of overlap with prior work. The reviewers also noted that the experimental results did not build a strong case for the proposed algorithm.

**Consistency Experiment:**

NeurIPS has a long history of experimentation. In 2014, NeurIPS ran an experiment in which 10% of submissions were reviewed by two independent committees to quantify the randomness in the review process. This year, we repeated a variant of this experiment to see how the quality of the review process has changed over time.  This paper was part of the experiment and was therefore assigned to two committees (consisting of reviewers, an Area Chair, and a Senior Area Chair) that reached independent decisions.  If both committees made the same recommendation, this recommendation was followed. If a single committee recommended acceptance, the paper was accepted (with the exception of a few cases in which the other committee identified what we considered a fatal flaw, e.g., an error in a key result).

This copy’s committee reached the following decision: **Reject**

The other committee assigned to the paper recommended **Accept (Poster)**.  You can find the other set of reviews, along with any follow up discussion with the authors here:
https://openreview.net/forum?id=90c-FVYJ5rL